# NEAR-OPTIMAL TRUST-AWARE MULTI-ARMED BANDITS

## ABSTRACT

Multi-armed bandit (MAB) algorithms have achieved significant success in sequential decision-making applications, under the premise that humans perfectly implement the recommended policy. However, existing methods often overlook the crucial factor of human trust in learning algorithms. When trust is lacking, humans may deviate from the recommended policy, leading to undesired learning performance. Motivated by this gap, we study the trust-aware MAB problem by integrating a dynamic trust model into the standard MAB framework. Specifically, it assumes that the recommended and actually implemented policy differs depending on human trust, which in turn evolves with the quality of the recommended policy. We establish the minimax regret in the presence of the trust issue and demonstrate the suboptimality of vanilla MAB algorithms such as the upper confidence bound (UCB) algorithm. To overcome this limitation, we introduce a novel two-stage trust-aware procedure that provably attains near-optimal statistical guarantees. A simulation study is conducted to illustrate the benefits of our proposed algorithm when dealing with the trust issue.

## 1 INTRODUCTION

The (stochastic) multi-armed bandit (MAB) algorithms have achieved remarkable success across diverse domains related to sequential decision-making, including healthcare intervention (Tewari & Murphy, 2017; Rabbi et al., 2018), recommendation systems (Li et al., 2010; Kallus & Udell, 2020), and dynamic pricing (Kleinberg & Leighton, 2003; Wang et al., 2021), to name a few. While a variety of prior work has been dedicated to this problem, existing results often emphasize learning uncertain environments and overlook humans' willingness to trust the output of learning algorithms, assuming humans implement the recommended policy with perfect precision. However, in numerous real-world applications, there may exist a discrepancy between the recommended and actually executed policy, which heavily relies on the humans' trust in the recommendations. The humans would embrace the suggested policy only if they have a high level of trust; otherwise, they tend to adhere to their own policy. Meanwhile, high-quality recommendations, in turn, can foster growing trust throughout the decision-making process. One concrete example can be found in human-robot interaction (HRI), where autonomous systems such as robots are employed to assist humans in performing tasks (Chen et al., 2020). Although the robot is fully capable of completing tasks, a novice user may not trust the robot and refuse its suggestion, leading to inefficient collaboration. Additionally, trust-aware decision-making finds crucial applications in emergency evacuations (Robinette & Howard, 2012). Myopic individuals might follow their own policy, such as the shortest path strategy, which is not necessarily optimal due to traffic congestion. If the evacuation plan designer fails to account for the uncertainty in action execution caused by trust issues, a skeptical individual is unlikely to follow the recommendations, potentially resulting in disastrous consequences.

This motivates research on trust-aware sequential decision-making, where deviations in decision implementation arising from trust issues need to be taken into account. Human trust should be monitored and influenced during the decision-making process in order to achieve optimal performance. Despite the foundational importance, studies of trust-aware sequential decision-making remain largely unexplored. Only recently have researchers begun to empirically incorporate trust into algorithm design (Moyano et al., 2014; Azevedo-Sa et al., 2020; Xu & Dudek, 2016; Chen et al., 2018; Akash et al., 2019). However, the theoretical understanding of trust-aware sequential decision-making still remains an open area of research.

In this paper, we take an initial step by formulating the problem of trust-aware MAB composed of three key components: (i) uncertain environments and unknown *human trust* that can be learned and influenced, (ii) a *policy-maker* that designs MAB algorithms for decision recommendation, and (iii) an *implementer* who may or may not execute the recommended action based on trust, which is unobservable to the policy-maker. The sequential decision-making process can be described as follows. At each time $h$, (1) the policy-maker selects an arm $a_h^{\mathsf{pm}}$ according to some policy $\pi_h^{\mathsf{pm}}$ and recommends it to the implementer that takes actions in reality; (2) the implementer pulls an arm $a_h^{\mathsf{ac}}$ according to some policy $\pi_h^{\mathsf{ac}}$, which depends on the recommended arm $a_h^{\mathsf{pm}}$, her own policy $\pi_h^{\mathsf{own}}$, and her trust $t_h$ in the policy-maker; (3) the implementer receives a random reward $R_h(a_h^{\mathsf{ac}})$ associated with the pulled arm $a_h^{\mathsf{ac}}$. The goal is to design a policy $\pi^{\mathsf{pm}}$ for the policy-maker to maximize the expected cumulative rewards received, that is, $\mathbb{E}[\sum_{h=1}^{H} R_h(a_h^{\mathsf{ac}})]$.

One critical issue surrounding the trust-aware MAB lies in *decision implementation deviation*. The policy-maker loses full control of exploration and exploitation due to the trust issue, which significantly hurts the balanced trade-off attained by the standard MAB methods. When the implementer's trust level is low, she will not explore the uncertain environment effectively as desired by the recommended policy $\pi^{\mathsf{pm}}$, while at the same time failing to fully exploit the optimal arm even after it has been identified. In particular, when her own policy $\pi^{\mathsf{own}}$ outperforms most arms, ignoring deviations in decision execution and blindly applying vanilla MAB algorithms, such as the upper confidence bound (UCB) algorithm, can severely decrease performance. Indeed, when $\pi^{\mathsf{own}}$ performs relatively well, the suboptimal arms are rarely selected by the implementer. The UCB-type algorithm, however, attempts to keep pulling those suboptimal arms for exploration purposes, causing a persistent decline in trust. This further exacerbates decision implementation deviation and hinders effective exploration, resulting in a large regret.

In light of such challenges, a fundamental question we seek to address in this paper is:

> *Can we design a trust-aware MAB algorithm that achieves (near-)minimax optimal statistical guarantees in the presence of trust issues?*

## 1.1 CONTRIBUTIONS

Encouragingly, we deliver an affirmative answer to the question above. We integrate a dynamic trust model in the standard MAB framework that characterizes sequential decision-making under uncertain environments with unknown human trust behavior. We establish that the minimax lower bound on the $H$-period regret scales as $\Omega(\sqrt{KH})$, where $K$ is the size of the arm set. Furthermore, we show that the standard UCB algorithm (Lai et al., 1985; Auer et al., 2002a) can incur a near-linear regret $\widetilde{\Omega}(H)$ when the trust issue is present. Here and throughout, we use the asymptotic notations $O(\cdot)$ (resp. $\widetilde{O}(\cdot)$) to represent upper bounds on the growth rate up to a constant (resp. logarithmic) factor, and $\Omega(\cdot)$ (resp. $\widetilde{\Omega}(\cdot)$) for lower bounds similarly.

To address this gap, we design a novel trust-aware UCB algorithm consisting of two stages. The approach first identifies the arms that ensure the implementer's trust in recommendations grows. It then conducts trust-aware exploration and exploitation, simultaneously optimizing decisions and building trust. Notably, our procedure operates in an adaptive manner without requiring any knowledge of the implementer's trust level, trust set, or own policy. We characterize the regret of our algorithm as $\widetilde{O}(\sqrt{KH} + K^4)$, which provably achieves the minimax regret up to a logarithmic factor for a wide range of horizon $H$.

To the best of our knowledge, this is the first theory to develop a trust-aware algorithm that provably achieves near-minimax optimal statistical efficiency. The major technical contribution lies in the delicate characterization of the interplay between the trust dynamics and decision-making.

## 1.2 RELATED WORK

**Multi-armed bandits.** Since the seminal work (Robbins, 1952), the MAB problem has been extensively studied, which is well-understood to a large extent. We refer readers to Bubeck et al. (2012); Slivkins et al. (2019); Lattimore & Szepesvári (2020) for a comprehensive overview. Numerous algorithms—including the UCB (Lai et al., 1985; Auer et al., 2002a), successive elimination (SE) (Even-Dar et al., 2006; Auer & Ortner, 2010), and Thompson sampling (TS) (Thompson, 1933;

Agrawal & Goyal, 2012)—have been developed that provably attain the minimax regret (Auer et al., 2002b; Gerchinovitz & Lattimore, 2016). Our trust-aware MAB formulation shares the same spirit as numerous variants of the MAB problem that are motivated by diverse practical constraints in real-world applications. A non-comprehensive list of examples includes MAB with safety constraints where actions must satisfy uncertain and cumulative or round-wise constraints (Amani et al., 2019; Pacchiano et al., 2021; Badanidiyuru et al., 2018; Liu et al., 2021), transfer/meta-learning for MAB where datasets collected from similar bandits are leveraged to improve learning performances (Cai et al., 2024; Lazaric et al., 2013; Cella et al., 2020; Kveton et al., 2021), risk-averse MAB where the objective is to minimize risk measures such as Conditional Value-at-Risk (CVaR) and Value-at-Risk (VaR) (Sani et al., 2012; Cassel et al., 2018; Wang et al., 2023), etc. It is worth noting that deviations in decision implementations have been explored in the context of MABs. For instance, the incentive-compatible MAB (Frazier et al., 2014; Mansour et al., 2015) studies the scenarios where deviations arise from the nature of recommendations—exploitative arms are more likely to be followed than exploratory ones. In contrast, our framework focuses on deviations driven by trust issues.

**Trust-aware decision-making.** Trust-aware decision-making is largely driven by human-robot interaction, where trust between humans and autonomous systems presents as a fundamental factor in realizing their full potential (Azevedo-Sa et al., 2020; Xu & Dudek, 2016; Chen et al., 2018; Akash et al., 2019; Bhat et al., 2022). Research on trust-aware decision-making can be broadly divided into reactive and proactive approaches. The reactive approach employs a predetermined scheme that specifies the policy-maker's behaviors when the human's trust falls outside an optimal range (Azevedo-Sa et al., 2020; Xu & Dudek, 2016). The formulation considered in the current paper belongs to the proactive approach, which integrates human trust into the design of decision-making algorithms. In addition to MABs, Markov decision processes (MDPs) and Partially Observable Markov decision processes (POMDPs) are also widely used in modeling the trust-aware decision-making process (Chen et al., 2018; 2020; Bhat et al., 2022; Akash et al., 2019). This proactive approach allows the learner to adapt its recommendations in response to human trust. However, these studies are predominantly empirical and lack a theoretical foundation.

## 1.3 NOTATION

For any $a, b \in \mathbb{R}$, we define $a \vee b := \max\{a, b\}$ and $a \wedge b := \min\{a, b\}$. For any finite set $\mathcal{A}$, we let $\Delta(\mathcal{A})$ be the probability simplex over $\mathcal{A}$. For any positive integer $K$, denote by $[K] := \{1, 2, \cdots, K\}$. We use $\mathbb{1}\{\cdot\}$ to represent the indicator function. For any distributions $P, Q$, $\mathsf{KL}(P\|Q)$ stands for the KL-divergence. For any $a \in \mathbb{R}$, denote by $\lfloor a \rfloor$ (resp. $\lceil a \rceil$) the largest (resp. smallset) integer that is no larger (resp. smaller) than $a$. For any two functions $f(n), g(n) > 0$, the notation $f(n) \lesssim g(n)$ (resp. $f(n) \gtrsim g(n)$) means that there exists a constant $C > 0$ such that $f(n) \leq Cg(n)$ (resp. $f(n) \geq Cg(n)$). The notation $f(n) \asymp g(n)$ means that $C_0 f(n) \leq g(n) \leq C_1 f(n)$ holds for some constants $C_0, C_1 > 0$. In addition, $f(n) = o(g(n))$ means that $\limsup_{n \to \infty} f(n)/g(n) = 0$, $f(n) \ll g(n)$ means that $f(n) \leq c_0 g(n)$ for some small constant $c_0 > 0$, and $f(n) \gg g(n)$ means that $f(n) \geq c_1 g(n)$ for some large constant $c_1 > 0$.

## 1.4 ORGANIZATION

The rest of the paper is organized as follows. Section 2 formulates the problem and introduces definitions and assumptions. Section 3 presents our theoretical findings and the analysis of main theories is presented in Section 4. The detailed proofs and technical lemmas are deferred to the appendix. Section 5 presents the numerical performance of the proposed algorithms. We conclude with a discussion of future directions in Section 6.

## 2 PROBLEM FORMULATION

**Trust-aware multi-armed bandit.** We study a (stochastic) $K$-armed bandit, described by a sequence of i.i.d. random variables $\big(R_h(1), \cdots, R_h(K)\big)_{h \geq 1}$, where $R_h(k)$ denotes the random reward generated by arm $k$ at time $h$. The rewards are assumed to be bounded in $[0, 1]$ for any $k \in [K]$ and $h \geq 1$. The expected reward associated with arm $k$ is denoted by $r(k) := \mathbb{E}[R_1(k)]$ for any $k \in [K]$. We denote $r^\star$ the maximum expected reward, i.e., $r^\star := \max_{k \in [K]} r(k)$. A policy $\pi = \{\pi_h\}_{h \geq 1}$ is a

collection of distributions over the arm set, where $\pi_h \in \Delta([K])$ specifies the (possibly randomized) arm selection at time $h$.

The MAB game operates as follows: at each time step $h$, the policy-maker recommends an arm $a_h^{\text{pm}}$ to the implementer. Based on $a_h^{\text{pm}}$, the implementer pulls arm $a_h^{\text{ac}}$ according to some policy $\pi_h^{\text{ac}}$ (which may differ from $a_h^{\text{pm}}$) and receives a random reward $R_h(a_h^{\text{ac}})$ yielded by the arm $a_h^{\text{ac}}$.

**Decision implementation deviation.** To characterize deviations in decision implementation, we focus on the disuse trust behavior model, which is widely recognized by empirical studies in human-robot interaction scenarios (Chen et al., 2020; Bhat et al., 2024).

Specifically, the implementer is assumed to have an own policy $\pi^{\text{own}}$, which is *unknown* to the policy-maker. Given the recommended action $a_h^{\text{pm}}$, the implementer chooses to either follow the instruction $a_h^{\text{pm}}$ or take action according to her own policy $\pi_h^{\text{own}}$, based on her trust level $t_h$ at time $h$. More concretely, we assume that

$$a_h^{\text{ac}} = \begin{cases} a_h^{\text{pm}}, & \text{if } \chi_h = 1; \\ a_h^{\text{own}}, & \text{if } \chi_h = 0. \end{cases} \tag{1}$$

Here, $\chi_h \overset{\text{ind.}}{\sim} \text{Bern}(t_h), h \geq 1$ is a sequence of independent Bernoulli random variables, where $t_h$ represents the implementer's trust level in the policy-maker at time $h$. Without loss of generality, the trust level $t_h$ is assumed to be bounded in $[0, 1]$. When $\chi_h = 1$, the implementer adopts the recommendation $a_h^{\text{pm}}$. Otherwise, she chooses an arm $a_h^{\text{own}}$ according to her own policy $\pi_h^{\text{own}}$. In other words, at time $h$, the implementer adopts the recommendation $a_h^{\text{pm}}$ with probability $t_h$, while reverting to her own policy $\pi_h^{\text{own}}$ with probability $1 - t_h$. Intuitively, a higher trust level indicates a greater tendency for the implementer to follow the policy-maker.

We assume the policy-maker has access to information on whether or not the implementer adopts the recommendations, i.e., $\{\chi_h\}_{h \geq 1}$, which is well motivated by numerous real-world applications in the human-robot interaction domain (see e.g., Bhat et al. (2022)).

**Trust update mechanism.** We incorporate a dynamic trust model to capture how recommendations influence the implementer's evolving trust. The trust level $t_h$ at time $h$ is assumed to satisfy

$$t_h = \frac{\alpha_h}{\alpha_h + \beta_h}, \tag{2}$$

where $\alpha_1 = \beta_1 = 1$, and

$$\alpha_h = \alpha_{h-1} + \mathbb{1}\{a_{h-1}^{\text{pm}} \in \mathcal{T}\}, \quad \text{and} \quad \beta_h = \beta_{h-1} + \mathbb{1}\{a_{h-1}^{\text{pm}} \notin \mathcal{T}\}, \quad \forall h \geq 2. \tag{3}$$

Here, the set $\mathcal{T} \subset [K]$ consists of arms that, when recommended, increase the implementer's trust in the policy-maker. Roughly speaking, the implementer's trust level is the frequency at which the recommended arm belongs to her trust set. The trust increases whenever an arm from the trust set $\mathcal{T}$ is recommended, resulting in a higher likelihood of following it in the future.

**Remark 1.** The trust update rule is motivated by the pattern observed in real-world human-subject experiments (Guo & Yang, 2021) and aligns with the well-known Laplace's rule of succession in probability theory. We note that the initial values $\alpha_1 = \beta_1 = 1$ are chosen for simplicity of presentation. Our results naturally extend to general cases with arbitrary constants $\alpha_1, \beta_1 > 0$.

We emphasize that the trust set $\mathcal{T}$ is chosen by the implementer and is *inaccessible* to the policy-maker. This information asymmetry is an inherent factor driving the challenges in such a hierarchical structure.

**Remark 2.** The trust set $\mathcal{T}$ can be viewed as representing the implementer's a priori beliefs or preferences regarding the arms. For instance, it may consist of arms that the implementer initially considers potentially optimal based on her limited knowledge. When the policy-maker suggests pulling an arm outside $\mathcal{T}$, it challenges the implementer's preconceptions and undermines her trust in the credibility of recommendations.

We denote by $\Pi(K, \mathcal{T}, \pi^{\text{own}})$ the class of trust-aware MABs that satisfy the conditions above.

To ensure effective learning when the policy-maker and implementer share aligned interests—both seeking to maximize rewards—we introduce a consistency assumption regarding the trust set $\mathcal{T}$, as described below.

**Assumption 1.** The trust set $\mathcal{T}$ includes at least an optimal arm, that is, $\exists k \in \mathcal{T}$ such that $r(k) = r^\star$.

Assumption 1 requires that the trust set contains the optimal arm when it is unique, and at least one optimal arm when multiple exist. All in all, it guarantees that the implementer's trust increases when an optimal arm is recommended, and hence building trust aligns with learning the optimal arm.

**Goal.** Our objective is to develop a policy $\pi^{\mathsf{pm}}$ for the policy-maker that minimizes the expected regret accumulated over $H$ time steps by the implementer policy $\pi^{\mathsf{ac}}$, namely,

$$\mathbb{E}[\mathsf{reg}_H(\pi^{\mathsf{pm}})] := \mathbb{E}\left[\sum_{h=1}^{H} \left(r^\star - r(a_h^{\mathsf{ac}})\right) \Big| a_h^{\mathsf{ac}} \sim \pi_h^{\mathsf{ac}}\right].$$

The expectation is taken with respect to the randomness of the executed policy $\pi^{\mathsf{ac}}$, which depends on the recommended policy $\pi^{\mathsf{pm}}$ and trust level $\{t_h\}_{h \geq 1}$. We emphasize that the policy $\pi_h^{\mathsf{pm}}$ at time $h$ depends only on observations anterior to $h$, specifically $\left\{\left(a_i^{\mathsf{ac}}, R_i(a_i^{\mathsf{ac}}), \chi_i\right)\right\}_{1 \leq i < h}$. Moreover, the own policy $\pi^{\mathsf{own}}$ and trust $\{t_i\}_{1 \leq i < h}$ is unobservable to the policy-maker.

## 3 MAIN RESULTS

### 3.1 SUB-OPTIMALITY OF UCB

One may naturally wonder whether we can resort to the classical MAB algorithms such as the UCB algorithm (Lai et al., 1985; Auer et al., 2002a) to solve the trust-aware MAB problem. Unfortunately, the answer is negative, which is formalized by the following regret lower bound for the UCB algorithm in Theorem 1 below. The proof can be found in Appendix B. For completeness of presentation, we present the UCB algorithm in Algorithm 2 in the appendix.

**Theorem 1.** *Suppose Assumption 1 holds. There exists a multi-armed bandit, a trust set, and an own policy such that the expected regret of the policy $\pi^{\mathsf{UCB}}$ generated by the UCB algorithm obeys*

$$\mathbb{E}[\mathsf{reg}_H(\pi^{\mathsf{UCB}})] \geq \frac{cH}{\sqrt{\log(H)}}, \tag{4}$$

*where $c$ is some constant independent of $H$.*

**Remark 3.** This theorem constructs a hard MAB instance with a fixed number of arms, emphasizing the suboptimality of horizon dependency $H$. It is straightforward to generalize it to encompass a broader range of cases.

Theorem 1 reveals the suboptimality of the UCB algorithm in the presence of the trust issue. To develop some intuition about its failure, note that the arm prescribed at time $h$ is only selected with probability $t_h$ due to deviations in decision implementation and that recommending an arm not in the trust set reduces the implementer's trust. As shall be clear from the analysis in Section 4, without accounting for the trust issue, the UCB algorithm adopts a relatively "aggressive" exploration strategy, causing the suboptimal arms to dominate the recommendations in the initial stage. As a consequence, the trust level $t_h$ delays rapidly to nearly zero within $o(H)$ time steps. This implies that the UCB algorithm effectively loses control of the decision-making in practice, with the implementer adhering to her own policy $\pi^{\mathsf{own}}$ for the remainder of the game. Therefore, the UCB algorithm incurs a near-linear regret $\widetilde{\Omega}(H)$ as long as $\pi^{\mathsf{own}}$ is not the optimal policy.

### 3.2 TRUST-AWARE UCB ALGORITHM

Revisiting the failure of the UCB approach highlights an important lesson for trust-aware policy design: maintaining a high level of trust is essential when exploring the suboptimal arms. In light of this observation, we proposed a two-stage trust-aware UCB paradigm that leverages the information contained in the pattern of decision implementation deviations. The first phase involves uniform arm selection by the policy-maker, aiming to identify the implementer's trust set and eliminate the arms outside this set. This procedure guarantees that the recommended policy will be followed subsequently. Once this elimination step is complete, the algorithm transitions to the second stage, where the policy-maker conducts trust-aware exploration and exploitation to identify the optimal arm. All in all, our strategy maintains the implementer's trust while effectively distinguishing the best arm.

---

**Algorithm 1** Trust-aware UCB

---

1: **Input:** arm set $[K]$, time horizon $H$.
2: Initialize arm set $\widehat{\mathcal{T}} \leftarrow \varnothing$, $m \leftarrow 30K^3 \log(H)$, and $H_0 \leftarrow 2mK$.
3: **for** $h = 1, \dots, H_0$ **do**
4:     Choose $a_h^{\mathsf{pm}} \leftarrow k$, where $k \leftarrow \lceil h/(2m) \rceil$.
5:     Observe $a_h^{\mathsf{ac}}, R_h(a_h^{\mathsf{ac}})$, and $\chi_h$.
6:     **if** $h \equiv 0 \mod 2m$ **then**
7:         Compute $Y_k^{(1)}$ and $Y_k^{(2)}$ as in (7).
8:         **if** $h = 2m$ **then**
9:             **if** $Y_1^{(1)} + Y_1^{(2)} \geq 1/2$ **then**
10:                 Update $\widehat{\mathcal{T}} \leftarrow \widehat{\mathcal{T}} \cup \{1\}$.
11:         **else**
12:             **if** $Y_k^{(2)} - Y_k^{(1)} \geq \lambda_k(\widehat{\mathcal{T}})$ (cf. (8)) **then**
13:                 Update $\widehat{\mathcal{T}} \leftarrow \widehat{\mathcal{T}} \cup \{k\}$.
14: Set $N_{H_0+1}^{\mathsf{ac}}(a) \leftarrow 1$ and $\mathsf{UCB}_{H_0+1}(a) \leftarrow 1$ for any $a \in [K]$.
15: **for** $h = H_0 + 1, \dots, H$ **do**
16:     Choose $a_h^{\mathsf{pm}} \leftarrow \arg\max_{a \in \widehat{\mathcal{T}}} \mathsf{UCB}_h(a)$ with ties broken uniformly at random.
17:     Observe $a_h^{\mathsf{ac}}, R_h(a_h^{\mathsf{ac}})$, and $\chi_h$.
18:     Update $N_{h+1}^{\mathsf{ac}}(a_h^{\mathsf{ac}})$, where

$$N_s^{\mathsf{ac}}(a) := 1 \vee \sum_{i=H_0+1}^{s-1} \mathbb{1}\{a_i^{\mathsf{ac}} = a\}, \quad \forall a \in [K], s > H_0 + 1. \tag{5}$$

19:     Update $\mathsf{UCB}_{h+1}(a_h^{\mathsf{ac}})$, where

$$\mathsf{UCB}_s(a) := 1 \wedge \left\{ \frac{1}{N_s^{\mathsf{ac}}(a)} \sum_{i=H_0+1}^{s-1} R_i(a) \mathbb{1}\{a_i^{\mathsf{ac}} = a\} + 2\sqrt{\frac{\log(H)}{N_s^{\mathsf{ac}}(a)}} \right\}, \quad \forall a \in [K], s > H_0 + 1. \tag{6}$$

20: **Output:** policy $\{a_h^{\mathsf{pm}}\}_{h \geq 1}$.

---

We summarize our trust-aware method in Algorithm 1 and elaborate on its two stages.

**Stage 1: trust set identification.**    As a preliminary stage for trust-aware exploration and exploitation, we aim to identify arms that do not belong to the implementer's trust set. The rationale behind this step is grounded in the trust update mechanism described in (2) and (3). Eliminating these arms allows the policy-maker to explore safely without losing the implementer's trust in Stage 2.

Since the trust level is not observable, we estimate it by counting the frequency with which each recommended arm is followed. Specifically, this phase runs for $K$ rounds and maintains an estimated trust set $\widehat{\mathcal{T}}$ at the end of each round, where the round length $m \asymp K^3 \log(H)$ is chosen to ensure accurate identification while controlling cumulative regret. In round $k$, Algorithm 1 selects the $k$-th arm $2m$ times and records whether the recommendations are followed. For each $k \in [K]$, we define

$$Y_k^{(1)} := \frac{1}{m} \sum_{i=1}^{m} \chi_{2m(k-1)+i} \quad \text{and} \quad Y_k^{(2)} := \frac{1}{m} \sum_{i=m+1}^{2m} \chi_{2m(k-1)+i}. \tag{7}$$

Moreover, we define the comparison threshold at round $k$ for each $k > 1$ as

$$\lambda_k(\widehat{\mathcal{T}}) = \begin{cases} 1/(5k), & \text{if } |\widehat{\mathcal{T}}| = 0; \\ -1/(5k), & \text{if } |\widehat{\mathcal{T}}| = k - 1; \\ 0, & \text{otherwise.} \end{cases} \tag{8}$$

For $k = 1$, we compare $Y_k^{(1)} + Y_k^{(2)}$ with $1/2$ to determine if arm 1 belongs to the implementer's trust set $\mathcal{T}$, as the sum approaches 1 if $1 \in \mathcal{T}$ and 0 otherwise with high probability. For $k > 1$,

we use $Y_k^{(2)} - Y_k^{(1)}$ to determine whether the $k$-th arm belongs to the trust set $\mathcal{T}$. This difference represents the discrepancy between the implementer's policy compliance frequency in the first and subsequent $m$ trials. It is not hard to see that the expectation of the difference $Y_k^{(2)} - Y_k^{(1)}$ exceeds $\lambda_k$ when $k \in \mathcal{T}$ and is smaller otherwise. Moreover, our choice of round length $m$ ensures that the observed difference aligns with its expectation with high probability.

**Stage 2: trust-aware exploration-exploitation.** Equipped with our estimate of the trust set $\widehat{\mathcal{T}}$, the remainder of our algorithm builds on the optimistic principle to distinguish the optimal arm. Thanks to the elimination procedure in Stage 1, the trust level $t_h$ is guaranteed to keep increasing in the second stage at a rate of $1 - t_h = O(1/h)$. This gradual increase ensures the implementer's trust remains high, allowing for effective exploration and exploitation, as will be demonstrated in the analysis in Section 4.

### 3.3 NEAR-MINIMAX OPTIMAL REGRET

We proceed to present the theoretical guarantees of Algorithm 1 in Theorem 2 below. The proof is postponed to Appendix C.

**Theorem 2.** *Suppose Assumption 1 holds. For any $K \geq 2$, the expected regret of the policy $\pi$ generated by Algorithm 1 satisfies*

$$\sup_{\Pi(K, \mathcal{T}, \pi^{\text{own}})} \mathbb{E}[\text{reg}_H(\pi)] \leq C_1 \sqrt{KH \log(H)} + C_2 K^4 \log^2(H), \tag{9}$$

*for some positive constants $C_1$ and $C_2$ independent of $H$ and $K$.*

The regret upper bound of our proposed algorithm contains two terms. The first term $O(\sqrt{KH \log(H)})$ represents the regret accumulated in the trust-aware exploration and exploitation phase, while the second term $O(K^4 \log^2(H))$ accounts for the trust set identification stage.

In addition, Theorem 3 below establishes the minimax lower bound on the regret of the trust-aware MAB problem. The proof can be found in Appendix D.

**Theorem 3.** *Suppose Assumption 1 holds. For any $K \geq 2$, one has*

$$\inf_{\pi} \sup_{\Pi(K, \mathcal{T}, \pi^{\text{own}})} \mathbb{E}[\text{reg}_H(\pi)] \geq c\sqrt{KH}, \tag{10}$$

*for some positive constant $c$ that is independent of $H$ and $K$.*

Here, the infimum is taken over the class of admissible policies obeying the policy at time $h$ depends only on observations prior to time $h$, that is, $\left(a_i^{\text{ac}}, R_i(a_i^{\text{ac}}), \chi_i\right)_{1 \leq i < h}$.

We provide several important implications as follows.

*(1) Near-minimax optimal regret.* When the time horizon $H$ is sufficiently large ($H \gtrsim K^7 \log^3(H)$), the second term becomes negligible. As a result, the regret upper bound (9) in Theorem 2 matches the minimax lower bound (10) in Theorem 3 up to a logarithmic term. This illustrates that near-optimal statistical efficiency can be attained despite the presence of trust issues. Also, in the classical setting where the recommended policy is executed exactly, the minimax lower bound on regret scales $\Omega(\sqrt{KH})$ (Auer et al., 2002b; Gerchinovitz & Lattimore, 2016). Therefore, the trust issue does not increase the complexity of the problem.

*(2) Adaptivity to unknown trust and own policy.* Our algorithm does not require any prior knowledge of the implementer's trust level $t$, trust set $\mathcal{T}$, or own policy $\pi^{\text{own}}$. This adaptivity ensures its broad applicability across diverse practical scenarios.

*(3) Cost of decision implementation deviation.* The term $\widetilde{O}(K^4)$ in the regret upper bound (9) can be seen as a burn-in cost incurred by deviations in arm selection. Before the policy-maker establishes enough trust, such deviations lead to extra regret if we do not impose any assumption on the own policy $\pi^{\text{own}}$.

**Remark 4.** While optimizing the dependency of burn-in cost on $K$ seems plausible, it remains unclear whether this term is an inherent consequence of the trust issue or an artifact of the algorithm

and proof. Therefore, we did not prioritize further optimization at this stage and focus on achieving the optimal dependence on $H$. We leave the task of addressing this burn-in cost and achieving the optimal dependence on $K$ to future work.

## 4 ANALYSIS

A critical difference in analyzing the trust-aware MAB compared to the standard MAB lies in capturing the trust behavior $\{t_h\}_{h\geq 1}$. For any policy $\pi^{\mathsf{pm}}$, the expected regret can be decomposed as

$$\mathbb{E}[\mathsf{reg}_H(\pi^{\mathsf{pm}})] = \mathbb{E}\underbrace{\left[\sum_{h=1}^{H} t_h\big(r^\star - r(a_h^{\mathsf{pm}})\big)\right]}_{=:\mathsf{reg}_H^{\mathsf{pm}}} + \mathbb{E}\underbrace{\left[\sum_{h=1}^{H}(1-t_h)(r^\star - r_h^{\mathsf{own}})\right]}_{=:\mathsf{reg}_H^{\mathsf{own}}}, \qquad (11)$$

where $r_h^{\mathsf{own}} := \mathbb{E}_{a\sim\pi_h^{\mathsf{own}}}[R_h(a)]$. Controlling $\mathsf{reg}_H^{\mathsf{pm}}$ is more or less standard where one can apply standard UCB-type arguments (see e.g. Lattimore & Szepesvári (2020)). On the other hand, controlling $\mathsf{reg}_H^{\mathsf{own}}$ requires more delicate effort due to the trust factor. In the classical MAB problem without the trust issue, $t_h = 1$ for any $h \geq 1$ and thus $\mathsf{reg}_H^{\mathsf{own}} = 0$. However, when the trust factor is incorporated, the trust level and arm selection are intertwined: the trust level influences the likelihood of the recommended arm being pulled, which in turn affects the trust level in the subsequent time step. Therefore, our main technical contribution towards developing regret bounds lies in pinning down the interaction among the trust dynamics $\{t_h\}_{h\geq 1}$, the recommended arms $\{a_h^{\mathsf{pm}}\}_{h\geq 1}$, and the selected arms $\{a_h^{\mathsf{ac}}\}_{h\geq 1}$.

### 4.1 REGRET LOWER BOUND FOR THE UCB ALGORITHM

We present the proof outline of Theorem 1 in this section. For the MAB instance, we set $K = 260$. The expected rewards are chosen to be $r(k) = 1/(6\sqrt{\log(H)+1})$ if $k < K$ and $r(k) = 1/(3\sqrt{\log(H)+1})$ if $k = K$. The trust set is set as $\mathcal{T} = \{K\}$ and the own policy $\pi^{\mathsf{own}}$ is chosen to be a uniform distribution over $\{K-1, K\}$, i.e., $\pi_h^{\mathsf{own}}(K-1) = \pi_h^{\mathsf{own}}(K) = 1/2$ for all $h \geq 1$. For convenience of notation, we define $\Delta_h^{\mathsf{own}} := r^\star - r_h^{\mathsf{own}} = 1/(12\sqrt{\log(H)+1})$.

Recall the decomposition in (11). Since $r^\star \geq r(a_h^{\mathsf{pm}})$ for any $h$, we can lower bound the regret

$$\mathsf{reg}_H \geq \mathsf{reg}_H^{\mathsf{own}} = \sum_{h=1}^{H} \Delta_h^{\mathsf{own}}(1 - t_h) = \frac{1}{12\sqrt{\log(H)+1}} \sum_{h=1}^{H}(1 - t_h). \qquad (12)$$

The key idea of the proof is to show that with high probability, the sum of the trust levels up to the final time step $\sum_{h=1}^{H} t_h$ is bounded by $O\big(\log^2(H)\big)$. Intuitively, the time horizon can be divided into three stages.

1. During the first stage $1 \leq h \leq \lfloor 256\log(H) \rfloor$, the trust level is simply upper bound by 1 and hence the sum is bounded by $h$.

2. In the second stage $\lfloor 256\log(H) \rfloor < h \leq \lfloor 1024\log(H) \rfloor$, as the suboptimal arms are insufficiently explored, their UCB estimates can be as high as that of the optimal arm. Consequently, the UCB algorithm recommends these suboptimal arms with at least a constant probability, causing a constant upper bound for the trust level ($1/3$ in our analysis). Therefore, the sum of trust levels in this stage scales $O(h/3)$.

3. In the third stage $\lfloor 1024\log(H) \rfloor < h \leq H$, due to the frequent recommendations of suboptimal arms in the previous two stages, the trust level keeps decaying at a rate of $O(1/h)$. Hence, the sum of trust levels in this step scales $\widetilde{O}(\log(h))$. Combining the trust accumulated in these three stages leads to the expression of $s_h$ in the analysis.

More specifically, let us define $s_h$ as follows:

$$s_h := \begin{cases} h, & \text{if } 1 \leq h \leq \lfloor 256\log(H) \rfloor; \\ \frac{1}{3}h + \frac{2}{3}\lfloor 256\log(H) \rfloor, & \text{if } \lfloor 256\log(H) \rfloor < h \leq \lfloor 1024\log(H) \rfloor; \\ \frac{1}{2}\lfloor 1024\log(H) \rfloor\left\{1 + \log\left(\frac{h}{\lfloor 1024\log(H) \rfloor}\right)\right\}, & \text{if } \lfloor 1024\log(H) \rfloor < h \leq H. \end{cases}$$
$$(13)$$

We shall show that with high probability, $\sum_{i=1}^{h} t_i \le s_h$ holds simultaneously for all $h \in [H]$. As an immediate consequence, we have $\mathbb{E}\big[\sum_{h=1}^{H} t_h\big] = o(H)$. Substituting this into (12) leads to the claimed conclusion.

## 4.2 Regret upper bound for our proposed algorithm

We proceed to provide the proof outline of Theorem 2 in this section. We shall bound the regret incurred in the two stages separately:

$$\mathrm{reg}_H = \underbrace{\sum_{h=1}^{H_0} \big(r^\star - r(a_h^{\mathsf{ac}})\big)}_{=:\mathrm{reg}_H^{\mathsf{ts}}} + \underbrace{\sum_{h=H_0+1}^{H} \big(r^\star - r(a_h^{\mathsf{ac}})\big)}_{=:\mathrm{reg}_H^{\mathsf{ta}}}. \tag{14}$$

**Stage 1: trust set identification.** In this stage, it suffices to upper bound the round length $m$ required to determine whether each arm $k$ belongs to the trust set $\mathcal{T}$. Towards this, recall the definitions of $Y_k^{(1)}$ and $Y_k^{(2)}$ in (7). For $k=1$, it is easy to see that $\mathbb{E}\big[Y_k^{(1)} + Y_k^{(2)}\big]$ equals $1 - \widetilde{O}(1/m)$ (resp. $\widetilde{O}(1/m)$) when $1 \in \mathcal{T}$ (resp. $1 \notin \mathcal{T}$). Moreover, the variance satisfies $\mathrm{Var}(Y_k^{(1)} + Y_k^{(2)}) = \widetilde{O}(1/m^2)$. Hence, applying the Bernstein inequality shows that with high probability, $Y_k^{(1)} + Y_k^{(2)}$ exceeds $1/2$ if $1 \in \mathcal{T}$, and falls below $1/2$ otherwise. As for $k > 1$, straightforward calculations yields $\mathbb{E}\big[Y_k^{(2)} - Y_k^{(1)}\big] = \Omega\big(1/(k^2 m) \vee S_{k-1}/k^2\big)$ and $\mathrm{Var}(Y_k^{(2)} - Y_k^{(1)}) = O\big((k - S_{k-1})/(k^2 m^2) \vee S_{k-1}/(mk^2)\big)$, where $S_{k-1} := \sum_{1 \le i \le k-1} \mathbb{1}\{i \in \mathcal{T}\}$. Given our choice of round length $m \asymp K^3 \log(H)$, we can invoke the Bernstein inequality to show that with high probability, $Y_k^{(2)} - Y_k^{(1)}$ exceeds the comparison threshold $\lambda_k$ in (8) when $k \in \mathcal{T}$. Similarly, one can also apply the same argument to show that the difference is smaller than $\lambda_k$ in the case $k \notin \mathcal{T}$. Combining these two observations allows us to reliably test whether $k \in \mathcal{T}$ for each $k \in [K]$.

Therefore, the regret incurred in Stage 1 can be bounded by

$$\mathbb{E}\big[\mathrm{reg}_H^{\mathsf{ts}}\big] = \mathbb{E}\Bigg[\sum_{h=1}^{H_0} \big(r^\star - r(a_h^{\mathsf{ac}})\big)\Bigg] \le H_0 = 2mK \lesssim K^4 \log(H). \tag{15}$$

**Stage 2: trust-aware exploration-exploitation.** In view of (11), we decompose the regret in this stage into the following two parts:

$$\mathrm{reg}_H^{\mathsf{ta}} = \underbrace{\sum_{h=H_0+1}^{H} t_h\big(r^\star - r(a_h^{\mathsf{pm}})\big)}_{\mathrm{reg}_H^{\mathsf{ta\text{-}pm}}} + \underbrace{\sum_{h=H_0+1}^{H} (1 - t_h)(r^\star - r_h^{\mathsf{own}})}_{\mathrm{reg}_H^{\mathsf{ta\text{-}own}}}. \tag{16}$$

To control the second term $\mathrm{reg}_H^{\mathsf{ta\text{-}own}}$, we note the key observation that $\widehat{\mathcal{T}}$ output by Stage 1 accurately estimate the trust set $\mathcal{T}$ with high probability. Consequently, one can use induction to show that the trust level $t_h$ obeys

$$1 - t_h = \left(1 - \frac{1}{1 + H_0 + h}\right)(1 - t_{h-1}) \lesssim \frac{H_0}{h} = \frac{K^4 \log(H)}{h}. \tag{17}$$

As a direct result, $\mathrm{reg}_H^{\mathsf{ta\text{-}own}}$ can be controlled by

$$\mathbb{E}\big[\mathrm{reg}_H^{\mathsf{ta\text{-}own}}\big] = \mathbb{E}\Bigg[\sum_{h=H_0+1}^{H} \Delta_h^{\mathsf{own}}(1 - t_h)\Bigg] \le \mathbb{E}\Bigg[\sum_{h=H_0+1}^{H} (1 - t_h)\Bigg] \lesssim K^4 \log^2(H). \tag{18}$$

Turning to $\mathrm{reg}_H^{\mathsf{ta\text{-}pm}}$, recall that $\chi_h = 1$ means that the implementer follows the recommended policy. One can express

$$\mathrm{reg}_H^{\mathsf{ta\text{-}pm}} = \sum_{h=H_0+1}^{H} t_h\big(r^\star - r(a_h^{\mathsf{pm}})\big) = \sum_{a \in \widehat{\mathcal{T}}} \Delta(a) \sum_{h=H_0+1}^{H} t_h \mathbb{1}\{a_h^{\mathsf{pm}} = a\}. \tag{19}$$

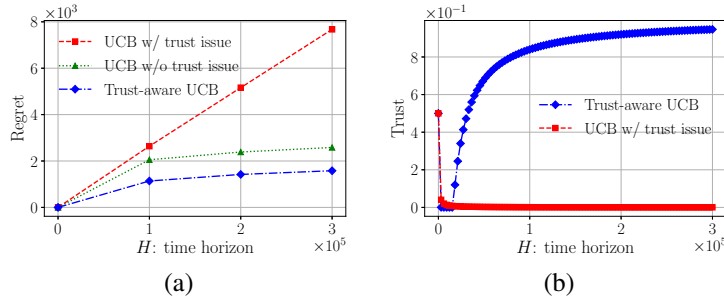

Figure 1: Comparison of the trust-aware and trust-blind algorithms: (a) regret; (b) trust level.

By invoking a concentration argument, we know that with high probability,

$$\sum_{h=H_0+1}^{H} \chi_h \mathbb{1}\{a_h^{\mathsf{pm}} = a\} \geq \sum_{h=H_0+1}^{H} t_h \mathbb{1}\{a_h^{\mathsf{pm}} = a\} - O\big(K^2 \log(H)\big).$$

Meanwhile, one can apply a standard UCB argument (see e.g., Lattimore & Szepesvári (2020)) to show that with high probability, $\sum_{h=H_0+1}^{H} \chi_h \mathbb{1}\{a_h^{\mathsf{pm}} = a\} \lesssim \frac{\log(H)}{\Delta^2(a)}$. Putting these two observations together leads to $\sum_{h=H_0+1}^{H} t_h \mathbb{1}\{a_h^{\mathsf{pm}} = a\} \lesssim O\big(\frac{\log(H)}{\Delta^2(a)} + K^2 \log(H)\big) \wedge H$. Combined with (19), this leads to

$$\mathbb{E}\big[\mathsf{reg}_H^{\mathsf{ta\text{-}pm}}\big] \lesssim \sqrt{KH \log(H)} + K^3 \log(H). \tag{20}$$

Combining (15), (18), and (20) completes the proof of Theorem 2.

## 5    NUMERICAL EXPERIMENTS

The numerical effectiveness of our algorithm is shown in Figure 1. Additional numerical experiments can be found in Appendix A. For the sake of comparison, we plot the regrets of the UCB algorithm both in the presence and absence of the trust issue. We set $K = 10$. For each arm $k$, the expected reward is set as $r(k) = k/(2K)$ and the random reward follows $R_h(k) \sim \mathcal{N}(r(k), 0.1)$. The implementer's trust set is set to be $\mathcal{T} = \{9, 10\}$ and the own policy $\pi_h^{\mathsf{own}}$ is chosen to be $\mathsf{Unif}(\{9, 10\})$ for all $h$.

Figure 1 (a) plots the regret vs. the horizon length $H$, with the regret averaged over 100 independent Monte Carlo trials; (b) depicts the trust level vs. the horizon length $H$ in a typical Monte Carlo trial. As can be seen, our trust-aware algorithm outperforms the (trust-blind) UCB algorithm in the presence of trust issues, whose regret grows linearly as predicted by our theorem. When recommendations are fully trusted and followed, our algorithm achieves comparable performances to the UCB algorithm. In addition, our algorithm maintains a high level of trust after the initial stage, whereas the trust level of the UCB algorithm decays rapidly to near zero, which is consistent with our theory.

## 6    DISCUSSION

We have studied the trust-aware MAB problem, where decision-making needs to account for deviations in decision implementation due to human trust issues. We established the suboptimality of vanilla MAB algorithms when faced with the trust issue and proposed a two-stage trust-aware algorithm, which achieves provable (near-)minimax optimal statistical guarantees.

Moving forward, several extensions are worth pursuing. To begin with, the proposed algorithm achieves the minimax regret when the time horizon is not too small. Investigating whether it is possible to achieve minimax optimality across the entire $H$ range would be valuable. Also, it is important to develop a general framework that accommodates a wider variety of trust models, making the approach more robust and broadly applicable. Finally, extending the MAB framework to study trust-aware reinforcement learning within the MDP framework would be of great interest.

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

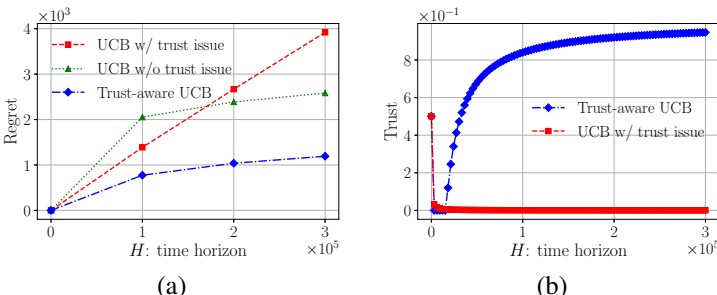

Figure 2: Comparison of the trust-aware and trust-blind algorithms where the own policy improves over time: (a) regret; (b) trust level.

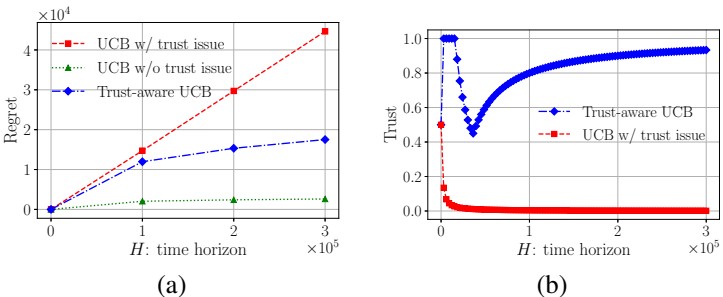

Figure 3: Comparison of the trust-aware and trust-blind algorithms where the trust set contains the best and highly suboptimal arms: (a) regret; (b) trust level.

## A  ADDITIONAL NUMERICAL EXPERIMENTS

In this section, we provide additional numerical experiments to support our theory. We set $K = 10$. For each arm $k$, the random reward follows $R_h(k) \sim \mathcal{N}(r(k), 0.1)$. The regret is averaged over 100 independent Monte Carlo trials, and the trust level is in a typical Monte Carlo trial.

Set the expected reward as $r(k) = k/(2K)$ for each $k \in [K]$. Figure 2 shows the case where the trust set $\mathcal{T} = \{9, 10\}$ contains near-optimal arms, with an improving own policy satisfying $\pi_h^{\mathsf{own}}(10) = 1/2 + h/(2H)$ and $\pi_h^{\mathsf{own}}(9) = 1/2 - h/(2H)$. Figure 3 demonstrates the scenario with $\mathcal{T} = \{1, 2, 3, 4, 10\}$, that is, the trust set consists of the best and highly suboptimal arm. The own policy obeys $\pi_h^{\mathsf{own}}(10) = 1/5 + 4h/(5H)$ and $\pi_h^{\mathsf{own}}(k) = 1/5 - h/(5H)$ for all $1 \le k \le 4$. In both these cases, our proposed algorithm outperforms the trust-blind UCB, which incurs linear regrets.

Figure 4 exhibits the case where the trust set contains the best-50% arms, i.e., $\mathcal{T} = \{k : 6 \le k \le 10\}$. The own policy is chosen to improve over time, with $\pi_h^{\mathsf{own}}(10) = 1/5 + 4h/(5H)$ and $\pi_h^{\mathsf{own}}(k) = 1/5 - h/(5H)$ for all $6 \le k < 10$. As can be seen, our algorithm achieves a regret performance comparable to that of the standard UCB algorithm, demonstrating a regret scaling of $\widetilde{O}(\sqrt{H})$. In this case, building trust is relatively easy as the trust set comprises many suboptimal arms. Consequently, the impact of trust issues is mild, and the regrets of the UCB algorithm with and without trust issues are nearly identical.

Finally, let us consider the case where multiple optimal arms exist and the trust set only contains one. We choose $r(k) = k/(2K)$ for each $k > 1$ and $r(1) = r(10)$. Figure 5 presents the case where the trust set is $\mathcal{T} = \{9, 10\}$ and own policy obeys $\pi_h^{\mathsf{own}}(10) = 1/2 + h/(2H)$ and $\pi_h^{\mathsf{own}}(9) = 1/2 - h/(2H)$. As evident from the plot, our proposed algorithm achieves a substantially smaller regret than the trust-blind UCB.

## B  PROOF OF THEOREM 1

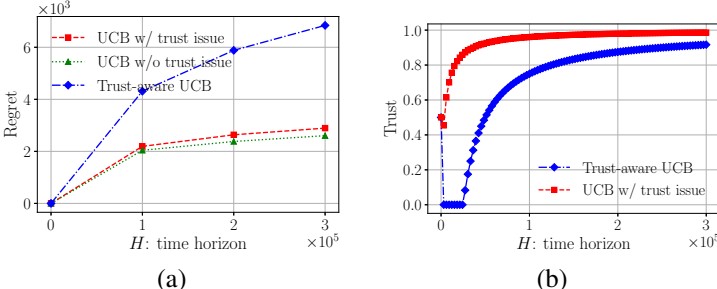

Figure 4: Comparison of the trust-aware and trust-blind algorithms where the trust set contains the best-50% arms: (a) regret; (b) trust level.

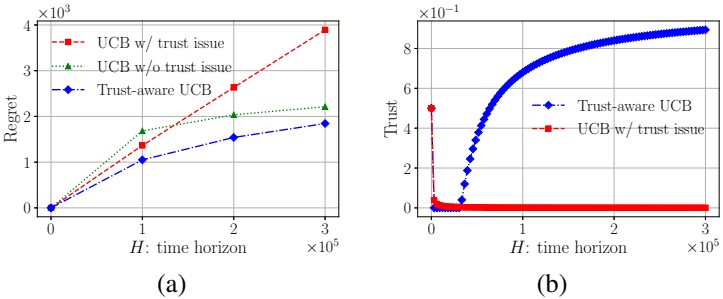

Figure 5: Comparison of the trust-aware and trust-blind algorithms where the trust set only contains one optimal arm: (a) regret; (b) trust level.

Recall the definition of random variables $\{\chi_i\}_{i \geq 1}$, where $\chi_i = 1$ indicates the implementer follows the recommended policy $\pi_i^{\mathsf{pm}}$ at time $i$, whereas $\chi_i = 0$ means the implementer takes her own policy $\pi^{\mathsf{own}}$. Let us define a sequence of random variables $\{\gamma_i\}_{i \geq 1}$, where $\gamma_i := \mathbb{1}\{a_i^{\mathsf{own}} = K\}$ indicates whether or not the implementer's own policy chooses the $K$-th arm at time $i$. In addition, we define the filtration $\mathcal{F}_i$ generated by the information by time $i$, that is

$$\mathcal{F}_i = \sigma\left(\left(a_j^{\mathsf{pm}}, a_j^{\mathsf{ac}}, R(a_j^{\mathsf{ac}}), \chi_j\right)_{j \in [i]}\right).$$

By Algorithm 1 and the trust update rule in Section 2, $\{t_i\}_{i \geq 1}$ and $\left\{\left(\mathsf{UCB}_i(k)\right)_{k \in [K]}\right\}_{i \geq 1}$ are predictable with respect to the filtration $\{\mathcal{F}_i\}_{i \geq 1}$, that is, $t_{i+1} \in \mathcal{F}_i$ and $\mathsf{UCB}_{i+1}(k) \in \mathcal{F}_i$ for all $k \in [K]$. Moreover, according to the assumption on the policy implementation deviation in Section 2, the distribution of $\chi_i$ conditional on the filtration $\mathcal{F}_{i-1}$ is given by

$$\chi_i \mid \mathcal{F}_{i-1} \sim \mathrm{Bern}(t_i).$$

Now let us begin the proof. As a reminder, the expected rewards of the MAB instance are set as

$$r(k) = \begin{cases} \dfrac{1}{6\sqrt{\log(H)+1}}, & \text{if } k < K; \\ \dfrac{1}{3\sqrt{\log(H)+1}}, & \text{if } k = K. \end{cases}$$

The implementer's own policy is chosen to be $\pi_h^{\mathsf{own}} \equiv \pi^{\mathsf{own}} = \mathsf{Unif}(\{K-1, K\})$ for all $h \geq 1$. As a result, we have $\Delta_h^{\mathsf{own}} := r^\star - r_h^{\mathsf{own}} = 1/(12\sqrt{\log(H)+1})$.

As $r^\star \geq r(a_h^{\mathsf{pm}})$ for any $h \geq 1$, we use the decomposition in (11) to lower bound the regret

$$\mathsf{reg}_H \geq \mathsf{reg}_H^{\mathsf{own}} = \Delta^{\mathsf{own}} \mathbb{E}\left[\sum_{h=1}^H (1 - t_h)\right] = \frac{1}{12\sqrt{\log(H)+1}}\left(H - \mathbb{E}\left[\sum_{h=1}^H t_h\right]\right). \tag{23}$$

---

**Algorithm 2** (Trust-blind) UCB

---

1: **Input:** arm set $[K]$, time horizon $H$.
2: Initialize arm set $\mathcal{A} \leftarrow [K]$. Set $N_1^{\mathsf{tb}}(a) \leftarrow 1$ and $\mathsf{UCB}_1^{\mathsf{tb}}(a) \leftarrow 1$ for any $a \in [K]$.
3: **for** $h = 1, \ldots, H$ **do**
4:      Choose $a_h^{\mathsf{pm}} \leftarrow \operatorname{argmax}_{a \in \mathcal{A}} \mathsf{UCB}_h^{\mathsf{tb}}(a)$ with ties broken uniformly at random.
5:      Observe $a_h^{\mathsf{ac}}$, $R_h(a_h^{\mathsf{ac}})$, and $\chi_h$.
6:      Update $N_{h+1}^{\mathsf{tb}}(a_h^{\mathsf{ac}})$, where

$$N_s^{\mathsf{tb}}(a) := 1 \vee \sum_{i=1}^{s-1} \mathbb{1}\{a_i^{\mathsf{ac}} = a\}, \quad \forall a \in [K], s > 1. \tag{21}$$

7:      Update $\mathsf{UCB}_{h+1}^{\mathsf{tb}}(a_h^{\mathsf{ac}})$, where

$$\mathsf{UCB}_s^{\mathsf{tb}}(a) := \frac{1}{N_s^{\mathsf{tb}}(a)} \sum_{i=1}^{s-1} R_i(a) \mathbb{1}\{a_i^{\mathsf{ac}} = a\} + 2\sqrt{\frac{\log(H)}{N_s^{\mathsf{tb}}(a)}}, \quad \forall a \in [K], s > 1. \tag{22}$$

8: **Output:** policy $\{a_h^{\mathsf{pm}}\}_{h \geq 1}$.

---

This implies it boils down to controlling the cumulative sum of the trust levels.

Towards this, recall the definition of $s_h$ in (13) where

$$s_h := \begin{cases} h, & \text{if } 1 \leq h \leq \lfloor 256 \log(H) \rfloor; \\ \dfrac{1}{3} h + \dfrac{2}{3} \lfloor 256 \log(H) \rfloor, & \text{if } \lfloor 256 \log(H) \rfloor < h \leq \lfloor 1024 \log(H) \rfloor; \\ \dfrac{1}{2} \lfloor 1024 \log(H) \rfloor \left\{ 1 + \log\left(\dfrac{h}{\lfloor 1024 \log(H) \rfloor}\right) \right\}, & \text{if } \lfloor 1024 \log(H) \rfloor < h \leq H. \end{cases}$$

Let us define the event

$$\mathcal{H}_h := \left\{ \sum_{i=1}^{h} t_i > s_h \right\}, \tag{24}$$

for each $h \geq 1$. We shall show that

$$\mathbb{P}\left( \bigcup_{i=1}^{H} \mathcal{H}_i \right) \leq 4H^{-1/3}. \tag{25}$$

As an immediate consequence, one obtains

$$\begin{aligned} \mathbb{E}\left[ \sum_{h=1}^{H} t_h \right] &\leq \mathbb{E}\left[ \mathbb{1}\{\mathcal{H}_H^{\mathsf{c}}\} \sum_{h=1}^{H} t_h \right] + \mathbb{E}\left[ \mathbb{1}\{\mathcal{H}_H\} \sum_{h=1}^{H} t_h \right] \\ &\leq s_H + \mathbb{P}(\mathcal{H}_H) H \\ &\leq 512\big(\log(H) + 1\big) + 4H^{2/3}, \end{aligned}$$

where the last step applies (13), (25) and the fact that $t_h \leq 1$ for any $h \geq 1$. Substituting this into (23) yields that for $H$ sufficiently large,

$$\mathsf{reg}_H \geq \frac{1}{12\sqrt{\log(H) + 1}} \Big( H - 512\big(\log(H) + 1\big) - 4H^{2/3} \Big) \geq c \frac{H}{\sqrt{\log(H)}},$$

where $c > 0$ is some universal constant.

Therefore, it suffices to establish (25). To this end, we need the following Freedman's inequality (Cesa-Bianchi & Lugosi, 2006, Lemma A.8), which is a generalization of the Bernstein inequality for a sum of bounded martingale difference sequences. (See also Freedman (1975)).

**Lemma 1** (Freedman's inequality). *Let $X_1, \ldots, X_n$ be a bounded martingale difference sequence with respect to a filtration $(\mathcal{F}_i)_{0 \leq i \leq n}$ and with $|X_i| \leq R$. Let $S_i := \sum_{j=1}^{i} X_j$ be the associated martingale. Denote the sum of the conditional variances by*

$$\Sigma_n^2 := \sum_{i=1}^{n} \mathbb{E}\big[X_i^2 \mid \mathcal{F}_{i-1}\big]. \tag{26}$$

*Then for any $\tau, \sigma^2 > 0$, one has*

$$\mathbb{P}\bigg\{ \max_{i \in [n]} S_i > \tau \text{ and } \Sigma_n^2 \leq \sigma^2 \bigg\} \leq \exp\left( -\frac{\tau^2}{2(\sigma^2 + R\tau/3)} \right). \tag{27}$$

*As a consequence, for any $0 < \delta < 1$, one further has*

$$\mathbb{P}\bigg\{ \max_{i \in [n]} S_i > 3R \log(1/\delta) \vee \sqrt{3\sigma^2 \log(1/\delta)} \text{ and } \Sigma_n^2 \leq \sigma^2 \bigg\} \leq \delta. \tag{28}$$

With Lemma 1 in place, we proceed to prove (25).

- We start with the case $1 \leq h \leq \lfloor 256 \log(H) \rfloor$. As the trust level $t_i$ is bounded in $[0, 1]$ for any $i \geq 1$, it is straightforward to bound

$$\sum_{i=1}^{h} t_i \leq h = s_h,$$

where we recall the definition of $s_h$ in (13) for $h \leq \lfloor 256 \log(H) \rfloor$. Therefore, we have

$$\mathbb{P}(\mathcal{H}_h) = 0, \tag{29}$$

for all $1 \leq h \leq \lfloor 256 \log(H) \rfloor$.

- Next, let us fix an arbitrary $\lfloor 256 \log(H) \rfloor < h \leq \lfloor 1024 \log(H) \rfloor$. We decompose the sum

$$\sum_{i=1}^{h} t_i = \sum_{i=1}^{\lfloor 256 \log(H) \rfloor} t_i + \sum_{i=\lfloor 256 \log(H) \rfloor + 1}^{h} t_i. \tag{30}$$

We shall show that the trust level $t_i$ is upper bounded by $1/3$ for all $i > \lfloor 256 \log(H) \rfloor$ with high probability. By construction, any arm in the complement of the support of $\pi^{\mathsf{own}}$ (namely, $\{1, 2, \ldots, K-2\}$) gets pulled only when the implementer follows the recommended policy, i.e., when $\chi_i = 1$. Therefore, the following bound holds for any $1 \leq i \leq \lfloor 1024 \log(H) \rfloor$,

$$\sum_{k=1}^{K-2} N_i^{\mathsf{tb}}(k) \leq \sum_{j=1}^{i} \chi_j \leq i \leq 1024 \log(H). \tag{31}$$

As a result, this indicates that there exist at least three arms $k_1, k_2, k_3 \in [K]$ satisfying

$$\max_{j \in [3]} N_i^{\mathsf{tb}}(k_j) \leq 4 \log(H).$$

This further implies that

$$\mathsf{UCB}_i^{\mathsf{tb}}(k_j) = \left( \widehat{\mu}_i(k_j) + 2\sqrt{\frac{\log(H)}{N_i^{\mathsf{tb}}(k_j)}} \right) \wedge 1 = 1.$$

Therefore, we find that $\{k_1, k_2, k_3\} \subset \operatorname{argmax}_{k \in [K]} \mathsf{UCB}_i(k)$ for any $1 \leq i \leq \lfloor 1024 \log(H) \rfloor$. On the other hand, note that the trust increases only if $a_i^{\mathsf{pm}} = K$ by the choice of the trust set. Combining this observation with the uniformly random tie-breaking in Algorithm 1, we know that

$$\mathbb{P}\{a_i^{\mathsf{pm}} = K \mid \mathcal{F}_{i-1}\} \leq \frac{1}{4}, \tag{32}$$

for all $1 \le i \le \lfloor 1024 \log(H) \rfloor$. Let us define $X_i := \mathbb{1}\{a_i^{\mathsf{pm}} = K\} - \mathbb{P}\{a_i^{\mathsf{pm}} = K \mid \mathcal{F}_{i-1}\}$ for each $1 \le i \le \lfloor 1024 \log(H) \rfloor$. Straightforward calculation reveals that $|X_i| \le 1$,

$$\mathbb{E}\big[X_i \mid \mathcal{F}_{i-1}\big] = 0,$$

and

$$\mathbb{E}\big[X_i^2 \mid \mathcal{F}_{i-1}\big] = \mathbb{P}\{a_i^{\mathsf{pm}} = K \mid \mathcal{F}_{i-1}\}\big(1 - \mathbb{P}\{a_i^{\mathsf{pm}} = K \mid \mathcal{F}_{i-1}\}\big) \le \frac{1}{4}.$$

Applying Lemma 1 with $\delta = H^{-4/3}$, $R = 1$, and $\sigma^2 = i/4$, we obtain that for any $1 \le i \le \lfloor 1024 \log(H) \rfloor$,

$$\mathbb{P}\left\{ \sum_{j=1}^{i} \mathbb{1}\{a_j^{\mathsf{pm}} = K\} > \sum_{j=1}^{i} \mathbb{P}\{a_j^{\mathsf{pm}} = K \mid \mathcal{F}_{i-1}\} + 4\log(H) \vee \sqrt{i \log(H)} \right\} \le H^{-4/3}. \tag{33}$$

Note that when $i \ge \lfloor 256 \log(H) \rfloor \ge 225 \log(H)$, we have

$$4\log(H) \vee \sqrt{i \log(H)} \le i\left(\frac{4}{225} \vee \sqrt{\frac{1}{225}}\right) = \frac{1}{15} i. \tag{34}$$

Therefore, let us define the event

$$\mathcal{J}_i := \left\{ \sum_{j=1}^{i} \mathbb{1}\{a_j^{\mathsf{pm}} = K\} > \frac{19}{60} i \right\}, \tag{35}$$

for each $\lfloor 256 \log(H) \rfloor \le i \le \lfloor 1024 \log(H) \rfloor$. Combining (32), (33), and (34), we have $\mathbb{P}(\mathcal{J}_i) \le H^{-\frac{4}{3}}$ for all $\lfloor 256 \log(H) \rfloor \le i \le \lfloor 1024 \log(H) \rfloor$. It follows from the union bound that

$$\mathbb{P}\left( \bigcup_{i=\lfloor 256 \log(H) \rfloor}^{\lfloor 1024 \log(H) \rfloor} \mathcal{J}_i \right) \le H^{-\frac{1}{3}}, \tag{36}$$

In what follows, we shall work on the event $\bigcap_{i=\lfloor 256 \log(H) \rfloor}^{\lfloor 1024 \log(H) \rfloor} \mathcal{J}_i^{\mathsf{c}}$. Recall the trust update rule in (2)–(3). The trust level $t_i$ at time $i$ admits the following expression:

$$t_i = \frac{1}{1+i} + \frac{1}{1+i} \sum_{j=1}^{i-1} \mathbb{1}\{a_j^{\mathsf{pm}} = K\}. \tag{37}$$

This allows us to derive that for any $\lfloor 256 \log(H) \rfloor < i \le \lfloor 1024 \log(H) \rfloor + 1$,

$$t_i \le \frac{1}{1+i} + \frac{1}{i-1} \sum_{j=1}^{i-1} \mathbb{1}\{a_j^{\mathsf{pm}} = K\} \le \frac{1}{1+i} + \frac{19}{60} \le \frac{1}{3}, \tag{38}$$

for sufficiently large $H$ (for instance, $\lfloor 256 \log(H) \rfloor \ge 591$). As a result, this demonstrates that

$$\sum_{i=1}^{h} t_i = \sum_{i=1}^{\lfloor 256 \log(H) \rfloor} t_i + \sum_{i=\lfloor 256 \log(H) \rfloor + 1}^{h} t_i \le \lfloor 256 \log(H) \rfloor + \frac{1}{3}\big(h - \lfloor 256 \log(H) \rfloor\big) = s_h,$$

where the last step follows from the definition of $s_h$ in (13). Consequently, this shows that

$$\bigcap_{i=\lfloor 256 \log(H) \rfloor}^{\lfloor 1024 \log(H) \rfloor} \mathcal{J}_i^{\mathsf{c}} \subset \mathcal{H}_H^{\mathsf{c}}.$$

Recognizing that this holds for an arbitrary $\lfloor 256 \log(H) \rfloor < h \le \lfloor 1024 \log(H) \rfloor$, we find that

$$\bigcap_{i=\lfloor 256 \log(H) \rfloor}^{\lfloor 1024 \log(H) \rfloor} \mathcal{J}_i^{\mathsf{c}} \subset \bigcap_{i=\lfloor 256 \log(H) \rfloor + 1}^{\lfloor 1024 \log(H) \rfloor} \mathcal{H}_i^{\mathsf{c}}.$$

Combining this with (36), we conclude that

$$\mathbb{P}\left(\bigcup_{i=\lfloor 256\log(H)\rfloor+1}^{\lfloor 1024\log(H)\rfloor} \mathcal{H}_i\right) \leq \mathbb{P}\left(\bigcup_{i=\lfloor 256\log(H)\rfloor}^{\lfloor 1024\log(H)\rfloor} \mathcal{J}_i\right) \leq H^{-\frac{1}{3}}. \tag{39}$$

- As for any $h > \lfloor 1024\log(H)\rfloor$, we shall prove

$$\mathbb{P}\left(\left(\bigcup_{i=\lfloor 256\log(H)\rfloor}^{\lfloor 1024\log(H)\rfloor} \mathcal{J}_i\right) \bigcup \left(\bigcup_{i=\lfloor 1024\log(H)\rfloor+1}^{h} \mathcal{H}_i\right)\right) \leq H^{-\frac{1}{3}} + 2hH^{-\frac{4}{3}}. \tag{40}$$

Then taking $h = H$ leads to

$$\mathbb{P}\left(\bigcup_{i=\lfloor 1024\log(H)\rfloor+1}^{H} \mathcal{H}_i\right) \leq H^{-\frac{1}{3}} + 2H \cdot H^{-\frac{4}{3}} = 3H^{-1/3}.$$

Towards this, we would like to establish (40) by induction. To begin with, (40) holds for the base case $h = \lfloor 1024\log(H)\rfloor$ as shown in (39).

Next, let us fix an arbitrary $h > \lfloor 1024\log(H)\rfloor$ and assume that (40) holds for $h$. We wish to prove the claim for $h+1$. By the trust update rule in (2) and (3), we can also express the trust level $t_i$ at time $i$ as

$$t_i = \left(1 - \frac{1}{1+i}\right)t_{i-1} + \frac{1}{1+i}\mathbb{1}\{a_{i-1}^{\mathsf{pm}} = K\}. \tag{41}$$

For each $i > \lfloor 1024\log(H)\rfloor$, let us define the event

$$\mathcal{J}_i := \left\{a_i^{\mathsf{pm}} = K\right\}. \tag{42}$$

On the event $\bigcap_{i=\lfloor 1024\log(H)\rfloor+1}^{h} \mathcal{J}_i^{\mathsf{c}}$, we can use (41) to derive

$$t_i = \left(1 - \frac{1}{1+i}\right)t_{i-1} = \left(1 - \frac{1}{1+i}\right)\left(1 - \frac{1}{i}\right)t_{i-2} = \frac{2 + \lfloor 1024\log(H)\rfloor}{1+i}t_{\lfloor 1024\log(H)\rfloor+1},$$

for all $\lfloor 1024\log(H)\rfloor < i \leq h+1$. Moreover, recall from (38) that on the event $\bigcap_{i=\lfloor 256\log(H)\rfloor}^{\lfloor 1024\log(H)\rfloor} \mathcal{J}_i^{\mathsf{c}}$, the trust at time $i$ satisfies $t_i \leq \frac{1}{3}$ for any $\lfloor 256\log(H)\rfloor < i \leq \lfloor 1024\log(H)\rfloor + 1$. This reveals that on the event $\bigcap_{i=\lfloor 256\log(H)\rfloor}^{h} \mathcal{J}_i^{\mathsf{c}}$, the trust level at time $i$ obeys

$$t_i = \frac{2 + \lfloor 1024\log(H)\rfloor}{1+i}t_{\lfloor 1024\log(H)\rfloor+1} \leq \frac{2 + \lfloor 1024\log(H)\rfloor}{3(1+i)} \leq \frac{\lfloor 1024\log(H)\rfloor}{2i}, \tag{43}$$

for all $\lfloor 1024\log(H)\rfloor < i \leq h+1$ provided $H$ is sufficiently large. As a consequence, on the event $\bigcap_{i=\lfloor 256\log(H)\rfloor}^{h} \mathcal{J}_i^{\mathsf{c}}$, the following holds for all $\lfloor 1024\log(H)\rfloor < i \leq h+1$:

$$\sum_{j=1}^{i} t_j = \sum_{j=1}^{\lfloor 256\log(H)\rfloor} t_j + \sum_{j=\lfloor 256\log(H)\rfloor+1}^{\lfloor 1024\log(H)\rfloor} t_j + \sum_{j=\lfloor 1024\log(H)\rfloor+1}^{i} t_j$$

$$\overset{(i)}{\leq} \lfloor 256\log(H)\rfloor + \frac{1}{3}\left(\lfloor 1024\log(H)\rfloor - \lfloor 256\log(H)\rfloor\right) + \sum_{j=\lfloor 1024\log(H)\rfloor+1}^{i} \frac{\lfloor 1024\log(H)\rfloor}{2j}$$

$$\overset{(ii)}{\leq} \frac{1}{2}\lfloor 1024\log(H)\rfloor + \int_{\lfloor 1024\log(H)\rfloor}^{i} \frac{\lfloor 1024\log(H)\rfloor}{2x}\,\mathrm{d}x$$

$$= \frac{1}{2}\lfloor 1024\log(H)\rfloor + \frac{1}{2}\lfloor 1024\log(H)\rfloor \log\left(\frac{i}{\lfloor 1024\log(H)\rfloor}\right) \overset{(iii)}{=} s_i.$$

Here, (i) arises from $t_i \in [0,1]$ for any $i \geq 1$, (38), and (43) ; (ii) is true as $4\lfloor x\rfloor \leq \lfloor 4x\rfloor$ for any $x \in [0,1]$; (iii) uses the definition of $s_h$ in (13). This demonstrates that

$$\bigcap_{i=\lfloor 256\log(H)\rfloor}^{h} \mathcal{J}_i^{\mathsf{c}} \subset \bigcap_{i=\lfloor 1024\log(H)\rfloor+1}^{h+1} \mathcal{H}_i^{\mathsf{c}}.$$

It follows that

$$\left(\bigcup_{i=\lfloor 256\log(H)\rfloor}^{\lfloor 1024\log(H)\rfloor} \mathcal{J}_i\right)\bigcup\left(\bigcup_{i=\lfloor 1024\log(H)\rfloor+1}^{h+1}\mathcal{H}_i\right)$$

$$\subset \left(\bigcup_{i=\lfloor 256\log(H)\rfloor}^{\lfloor 1024\log(H)\rfloor} \mathcal{J}_i\right)\bigcup\left(\bigcup_{i=\lfloor 256\log(H)\rfloor}^{h}\mathcal{J}_i\right)$$

$$= \left(\bigcup_{i=\lfloor 256\log(H)\rfloor}^{\lfloor 1024\log(H)\rfloor} \mathcal{J}_i\right)\bigcup\left(\bigcup_{i=\lfloor 1024\log(H)\rfloor+1}^{h}(\mathcal{J}_i\cap\mathcal{H}_i)\right)\bigcup\left(\bigcup_{i=\lfloor 1024\log(H)\rfloor+1}^{h}(\mathcal{J}_i\cap\mathcal{H}_i^{\mathsf{c}})\right)$$

$$\subset \left(\bigcup_{i=\lfloor 256\log(H)\rfloor}^{\lfloor 1024\log(H)\rfloor} \mathcal{J}_i\right)\bigcup\left(\bigcup_{i=\lfloor 1024\log(H)\rfloor+1}^{h}\mathcal{H}_i\right)\bigcup\left(\bigcup_{i=\lfloor 1024\log(H)\rfloor+1}^{h}(\mathcal{J}_i\cap\mathcal{H}_i^{\mathsf{c}})\right),$$

which leads to

$$\mathbb{P}\left(\left(\bigcup_{i=\lfloor 256\log(H)\rfloor}^{\lfloor 1024\log(H)\rfloor} \mathcal{J}_i\right)\bigcup\left(\bigcup_{i=\lfloor 1024\log(H)\rfloor+1}^{h+1}\mathcal{H}_i\right)\right)$$

$$\leq \mathbb{P}\left(\left(\bigcup_{i=\lfloor 256\log(H)\rfloor+1}^{\lfloor 1024\log(H)\rfloor} \mathcal{J}_i\right)\bigcup\left(\bigcup_{i=\lfloor 1024\log(H)\rfloor+1}^{h}\mathcal{H}_i\right)\right) + \sum_{i=\lfloor 1024\log(H)\rfloor+1}^{h}\mathbb{P}\left(\mathcal{J}_i\cap\mathcal{H}_i^{\mathsf{c}}\right).$$

We claim that for any $\lfloor 1024\log(H)\rfloor < i \leq h$, one has

$$\mathbb{P}\left(\mathcal{J}_i\cap\mathcal{H}_i^{\mathsf{c}}\right) \leq 2H^{-\frac{7}{3}}. \tag{44}$$

Assuming the validity of (44), we arrive at

$$\mathbb{P}\left(\left(\bigcup_{i=\lfloor 256\log(H)\rfloor}^{\lfloor 1024\log(H)\rfloor} \mathcal{J}_i\right)\bigcup\left(\bigcup_{i=\lfloor 1024\log(H)\rfloor+1}^{h+1}\mathcal{H}_i\right)\right) \leq H^{-\frac{1}{3}} + 2hH^{-\frac{4}{3}} + 2hH^{-\frac{7}{3}} \leq H^{-\frac{1}{3}} + 2(h+1)H^{-\frac{4}{3}},$$

leading to the claim in (40) for $h+1$. This completes the proof of the claim in (40) by standard induction arguments.

Therefore, it remains to prove (44). Before proceeding, we summarize several bounds for $s_h$ (cf. (13)) that will be useful for the proof. First of all, for any $i > \lfloor 1024\log(H)\rfloor$, one has

$$s_i = \frac{1}{2}\lfloor 1024\log(H)\rfloor\left(1+\log\left(\frac{i}{\lfloor 1024\log(H)\rfloor}\right)\right) \leq \frac{1}{2}\lfloor 1024\log(H)\rfloor\frac{i}{\lfloor 1024\log(H)\rfloor} = \frac{i}{2}, \tag{45}$$

where the inequality holds since $1 + \log(x) \leq x$ for any $x > 0$. Next, it is straightforward to check that (45) also holds for $i = \lfloor 1024\log(H)\rfloor$, namely, $s_{\lfloor 1024\log(H)\rfloor} = \frac{1}{3}\lfloor 1024\log(H)\rfloor + \frac{2}{3}\lfloor 256\log(H)\rfloor \leq \frac{1}{2}\lfloor 1024\log(H)\rfloor$. In addition, for any $i \geq \lceil 1024\log(H)\rceil$, we have

$$s_i \geq \frac{1}{2}\lfloor 1024\log(H)\rfloor \geq 511\log(H), \tag{46}$$

for sufficiently large $H$, and

$$s_i < \frac{1}{2}\lfloor 1024\log(H)\rfloor\left(1+\log(H)\right) \leq 512\log(H)\left(1+\log(H)\right). \tag{47}$$

With these results in place, let us begin proving (44). For each $i \geq 1$, let us define the random variables

$$X_i := \chi_i - t_i \quad \text{and} \quad Y_i := \gamma_i(1-\chi_i) - \frac{1}{2}(1-t_i),$$

where we recall $\gamma_i := \mathbb{1}\{a_i^{\mathsf{own}} = K\}$. In addition, we define the events

$$\mathcal{K}_i := \left\{ \sum_{j=1}^{i} \chi_j > \sum_{j=1}^{i} t_j + 7\log(H) \vee \sqrt{7\log(H)s_i} \right\},$$

$$\mathcal{L}_i := \left\{ \sum_{j=1}^{i} \gamma_j(1-\chi_j) < \sum_{j=1}^{i} \frac{1}{2}(1-t_j) - 7\log(H) \vee \sqrt{7\log(H)i} \right\},$$

for each $\lfloor 1024\log(H)\rfloor < i \leq h$. It is straightforward to check that $\mathbb{E}[X_i \mid \mathcal{F}_{i-1}] = \mathbb{E}[Y_i \mid \mathcal{F}_{i-1}] = 0$,

$$\mathbb{E}\left[X_i^2 \mid \mathcal{F}_{i-1}\right] = t_i(1-t_i) \leq t_i,$$
$$\mathbb{E}\left[Y_i^2 \mid \mathcal{F}_{i-1}\right] \leq 1,$$

for any $i \geq 1$. Invoking Lemma 1 with $\delta = H^{-\frac{7}{3}}$, $R = 1$, and $\sigma^2 = s_i$, we know that for each $\lfloor 1024\log(H)\rfloor < i \leq h$,

$$\mathbb{P}\{\mathcal{K}_i \cap \mathcal{H}_i^{\mathsf{c}}\} \leq H^{-7/3}, \tag{48}$$

and

$$\mathbb{P}\{\mathcal{L}_i \cap \mathcal{H}_i^{\mathsf{c}}\} \leq H^{-7/3}. \tag{49}$$

Recall the definition of $\mathcal{H}_i$ in (24) and $\pi_h^{\mathsf{own}} = \mathsf{Unif}(\{K-1, K\})$ for all $h \geq 1$. On the event $\mathcal{K}_i^{\mathsf{c}} \bigcap \mathcal{H}_i^{\mathsf{c}}$, we have

$$\sum_{k=1}^{K-2} N_i^{\mathsf{tb}}(k) \leq \sum_{j=1}^{i} \chi_j \leq \sum_{j=1}^{i} t_j + 7\log(H) \vee \sqrt{7\log(H)s_i}$$

$$\leq \left(1 + \frac{7}{511} \vee \sqrt{\frac{7}{511}}\right)s_i \leq \frac{3}{2}s_i.$$

where the last line uses the fact that $\sum_{j=1}^{i} t_j \leq s_i$ on the event $\mathcal{H}_i^{\mathsf{c}}$ and the lower bound of $s_i$ in (46). In particular, this allows us to obtain

$$\min_{1 \leq k \leq K-2} N_i^{\mathsf{tb}}(k) \leq \frac{1}{K-2} \sum_{k=1}^{K-2} N_i^{\mathsf{tb}}(k) \leq \frac{1}{258}\frac{3}{2}s_i = \frac{1}{172}s_i. \tag{50}$$

Meanwhile, on the event $\mathcal{L}_i^{\mathsf{c}} \cap \mathcal{H}_i^{\mathsf{c}}$, we can lower bound

$$N_i^{\mathsf{tb}}(K) \geq \sum_{j=1}^{i} \gamma_j(1-\chi_j) \geq \sum_{j=1}^{i} \frac{1}{2}(1-t_j) - 7\log(H) \vee \sqrt{7\log(H)i}$$

$$\geq \left(\frac{1}{2} - \frac{7}{511} \vee \sqrt{\frac{7}{511}}\right)i - \frac{1}{2}s_i \geq \frac{1}{6}s_i,$$

where the penultimate inequality holds because $\sum_{j=1}^{i} t_j \leq s_i$ on the event $\mathcal{H}_i^{\mathsf{c}}$ and $\log(H) \leq \frac{1}{511}s_i \leq \frac{1}{1022}i$ due to (45) and (46); the last inequality holds because of (45). This implies that

$$\mathsf{UCB}_i(K) \leq \left(r(K) + 4\sqrt{\frac{\log(H)}{N_i^{\mathsf{tb}}(K)}}\right) \wedge 1 \leq \left(r(K) + \sqrt{\frac{96\log(H)}{s_i}}\right) \wedge 1.$$

On the other hand, (50) allows us to derive

$$\max_{1 \leq k \leq K-2} \mathsf{UCB}_i(k) \geq 4\sqrt{\frac{\log(H)}{\min_{1 \leq k \leq K-2} N_i^{\mathsf{tb}}(K)}} \wedge 1 \geq \sqrt{\frac{2752\log(H)}{s_i}} \wedge 1. \tag{51}$$

By our construction of the instance, the reward of the optimal arm satisfies

$$r(K) = \frac{1}{3}\sqrt{\frac{1}{1 + \log(H)}} < \sqrt{\frac{512\log(H)}{9s_i}},$$

where the last step follows from (47). Taken collectively with the fact that $\sqrt{512/9} + \sqrt{96} < \sqrt{2752}$, this leads to

$$\mathsf{UCB}_i(K) < \max_{1 \leq k \leq K-2} \mathsf{UCB}_i(k).$$

By the arm selection procedure of the UCB algorithm, we know that $r(a_i^{\mathsf{pm}}) \neq K$. Recall the definition of $\mathcal{J}_i$ for $i > \lfloor 1024\log(H) \rfloor$ in (42). This implies that

$$\mathcal{K}_i^{\mathsf{c}} \cap \mathcal{L}_i^{\mathsf{c}} \cap \mathcal{H}_i^{\mathsf{c}} \subset \mathcal{J}_i^{\mathsf{c}} \cap \mathcal{H}_i^{\mathsf{c}},$$

which further yields

$$\mathcal{J}_i \cap \mathcal{H}_i^{\mathsf{c}} \subset \left(\mathcal{K}_i \cap \mathcal{H}_i^{\mathsf{c}}\right) \cup \left(\mathcal{L}_i \cap \mathcal{H}_i^{\mathsf{c}}\right).$$

As a result, combining (48), (49) with the union bound proves (44).

- Finally, putting (29), (39), and (40) collectively with the union bound establishes (25). This concludes the proof of Theorem 1.

## C  PROOF OF THEOREM 2

**Stage 1: trust set identification.**  Recall the definitions of $Y_k^{(1)}$ and $Y_k^{(2)}$ in (7):

$$Y_k^{(1)} := \frac{1}{m}\sum_{i=1}^{m}\chi_{2m(k-1)+i} \quad \text{and} \quad Y_k^{(2)} := \frac{1}{m}\sum_{i=m+1}^{2m}\chi_{2m(k-1)+i}.$$

In addition, we denote

$$\delta_k := Y_k^{(2)} - Y_k^{(1)}, \quad \forall k > 1.$$

As a reminder, for any $k > 1$, the value of $\delta_k$ will be used to test whether the $k$-th arm belongs to the implementer's trust set $\mathcal{T}$ or, equivalently, leads to increasing trust.

To this end, denote $S_k := \sum_{\ell=1}^{k} \mathbb{1}\{l \in \mathcal{T}\}$ for each $k \in [K]$ and we set $S_0 := 0$ by default. It is straightforward to see that for any $k \in [K]$ and $i \in [2m]$,

$$t_{2m(k-1)+i} = \mathbb{E}\big[\chi_{2m(k-1)+i}\big] = \begin{cases} \dfrac{1 + 2mS_{k-1} + i - 1}{2 + 2m(k-1) + i - 1}, & \text{if } k \in \mathcal{T}; \\[2mm] \dfrac{1 + 2mS_{k-1}}{2 + 2m(k-1) + i - 1}, & \text{if } k \notin \mathcal{T}. \end{cases}$$

- We begin with the case $k = 1$. Let us denote $Y_1 := Y_1^{(1)} + Y_1^{(2)}$.

  - If $1 \notin \mathcal{T}$, the expectation of $Y_1$ can be upper bounded by

  $$\mathbb{E}[Y_1] = \frac{1}{2m}\sum_{i=1}^{2m}\frac{1}{1+i} \leq \frac{1}{2m}\int_0^{2m}\frac{1}{1+x}\,\mathrm{d}x \leq \frac{1}{2m}\log(1 + 2m).$$

  As for the variance, we can compute

  $$V_1 := \sum_{i=1}^{2m}\mathrm{Var}(\chi_i) = \sum_{i=1}^{2m}t_i(1 - t_i) = \sum_{i=1}^{2m}\frac{i}{(1+i)^2}.$$

As $x \mapsto x/(a+x)^2$ is increasing in $[0, a]$ and decreasing in $[a, \infty)$ for any $a > 0$, we can bound

$$V_1 = \sum_{i=1}^{2m} \frac{i}{(1+i)^2} = \frac{1}{4} + \sum_{i=2}^{2m} \frac{i}{(1+i)^2} \leq \frac{1}{4} + \int_1^{2m} \frac{x}{(1+x)^2} \, \mathrm{d}x$$

$$= \frac{1}{1+2m} - \frac{1}{4} + \log\left(m + \frac{1}{2}\right) \leq \log(m),$$

where the last step holds as long as $m \geq 4$. Therefore, applying the Bernstein inequality shows that with probability at least $1 - H^{-2}$,

$$Y_1 \leq \mathbb{E}[Y_1] + \frac{4}{3} \frac{\log(H)}{m} + \frac{2}{m} \sqrt{V_1 \log(H)}$$

$$\leq \frac{\log(1+2m)}{2m} + \frac{4\log(H)}{3m} + \frac{2}{m} \sqrt{\log(m)\log(H)} \leq \frac{1}{2},$$

where the last step holds provided $m \geq 3\log(H)$ and $H$ is sufficiently large.

– On the other hand, if $1 \in \mathcal{T}$, we can compute

$$\mathbb{E}[Y_1] = \frac{1}{2m} \sum_{i=1}^{2m} \frac{i}{1+i} \geq 1 - \frac{1}{2m} \log(1+2m).$$

and

$$V_1 := \sum_{i=1}^{2m} \mathrm{Var}(\chi_i) = \sum_{i=1}^{2m} \frac{i}{(1+i)^2} \leq \log(m).$$

Invoking the Bernstein inequality yields that with probability exceeding $1 - H^{-2}$,

$$Y_1 \geq \mathbb{E}[Y_1] - \frac{4}{3} \frac{\log(H)}{m} - \frac{2}{m} \sqrt{V_1 \log(H)}$$

$$\geq 1 - \frac{\log(1+2m)}{2m} - \frac{4\log(H)}{3m} - \frac{2}{m} \sqrt{\log(m)\log(H)} \geq \frac{1}{2},$$

where the last step is true as long as $m \geq 3\log(H)$ and $H$ is sufficiently large.

– As a result, our procedure that adding arm 1 to the estimated trust set if $Y_1 \geq \frac{1}{2}$ correctly identifies $1 \in \mathcal{T}$ with probability at least $1 - H^{-2}$.

• We proceed to consider the case $k > 1$, where $\delta_k$ is used to identify the trust set.

– Let us first consider the case $k \notin \mathcal{T}$.
To begin with, we can apply the trust update mechanism to control the expectation of $\delta_k$ by

$$\mathbb{E}[\delta_k] = \frac{1}{m} \sum_{i=0}^{m-1} \frac{1 + 2S_{k-1}m}{2 + 2(k-1)m + m + i} - \frac{1}{m} \sum_{i=0}^{m-1} \frac{1 + 2S_{k-1}m}{2 + 2(k-1)m + i}$$

$$= -\sum_{i=0}^{m-1} \frac{1 + 2S_{k-1}m}{\big(2 + 2(k-1)m + i\big)\big(2 + 2(k-1)m + m + i\big)}$$

$$\leq -\int_0^m \frac{1 + 2S_{k-1}m}{\big(2 + 2(k-1)m + x\big)\big(2 + 2(k-1)m + m + x\big)} \, \mathrm{d}x$$

$$= -\frac{1}{m}(1 + 2S_{k-1}m) \log\left(1 + \frac{m^2}{4\big(1 + (k-1)m\big)(1 + km)}\right).$$

As $\log(1+x) \geq x/2$ for any $x \in [0, 1]$, one can further upper bound

$$\mathbb{E}[\delta_k] \leq -\frac{m(1 + 2S_{k-1}m)}{8\big(1 + (k-1)m\big)(1 + km)} \leq -\frac{1 + 2S_{k-1}m}{9(k-1)km}, \tag{52}$$

where the last step is true provided $m \gg 1$.

As for the variance of $\delta_k$, it is not hard to see that

$$V_k := \sum_{i=1}^{2m} \mathrm{Var}\left(\chi_{2(k-1)m+i}\right) = \sum_{i=1}^{2m} t_{2(k-1)m+i}(1 - t_{2(k-1)m+i})$$

$$= t_{2(k-1)m+1}\left(1 - t_{2(k-1)m+1}\right) + (1 + 2S_{k-1}m) \sum_{i=1}^{2m-1} \frac{1 + 2(k-1-S_{k-1})m + i}{\left(2 + 2(k-1)m + i\right)^2}.$$

Straightforward calculation yields

$$\sum_{i=1}^{2m-1} \frac{1 + 2(k-1-S_{k-1})m + i}{\left(2 + 2(k-1)m + i\right)^2}$$

$$\leq \int_0^{2m-1} \frac{1 + 2(k-1-S_{k-1})m}{\left(2 + 2(k-1)m + x\right)^2}\, \mathrm{d}x + \int_1^{2m} \frac{x}{\left(2 + 2(k-1)m + x\right)^2}\, \mathrm{d}x$$

$$\leq \int_0^{2m} \frac{1 + 2(k-1-S_{k-1})m + x}{\left(2 + 2(k-1)m + x\right)^2}\, \mathrm{d}x$$

$$= -\frac{1 + 2S_{k-1}m}{2 + 2km}\frac{m}{1 + (k-1)m} + \log\left(1 + \frac{m}{1 + (k-1)m}\right)$$

$$\leq -\frac{1 + 2S_{k-1}m}{2 + 2km}\frac{m}{1 + (k-1)m} + \frac{m}{1 + (k-1)m}$$

$$= \frac{m}{1 + (k-1)m}\frac{1 + 2(k-S_{k-1})m}{2(1 + km)},$$

where we use that fact that $x \mapsto x/(a + x)^2$ is increasing in $[0, a]$ and decreasing in $[a, \infty)$ for any $a > 0$ and $2 + 2(k-1)m > 2m$ for $k > 1$; the last inequality follows from $\log(1 + x) \leq x$ for any $x \geq 0$. Therefore, we find that

$$V_k \leq t_{2(k-1)m+1}\left(1 - t_{2(k-1)m+1}\right) + \frac{m(1 + 2S_{k-1}m)\left(1 + 2(k - S_{k-1})m\right)}{2\left(1 + (k-1)m\right)(1 + km)}$$

$$\leq \frac{1}{4} + \frac{2(1 + 2S_{k-1}m)(k - S_{k-1})}{(k-1)k}, \tag{53}$$

where the last line holds because $t_h(1 - t_h) \leq 1/4$ for any $h \geq 1$, $k - S_{k-1} \geq 1$, and $m \gg 1$.

Putting (52) and (53) together, we now invoke the Bernstein inequality to find that with probability at least $1 - H^{-2}$,

$$\delta_k \leq \mathbb{E}[\delta_k] + \frac{4}{3}\frac{\log(H)}{m} + \frac{2}{m}\sqrt{V_k \log(H)}$$

$$\leq -\frac{1 + 2S_{k-1}m}{9(k-1)km} + \frac{4}{3}\frac{\log(H)}{m} + \frac{2}{m}\sqrt{\frac{\log(H)}{4} + \frac{2(1 + 2S_{k-1}m)(k - S_{k-1})\log(H)}{(k-1)k}}$$

$$\leq -\frac{1 + 2S_{k-1}m}{9(k-1)km} + \frac{7}{3}\frac{\log(H)}{m} + \frac{2\sqrt{2}}{m}\sqrt{\frac{(k - S_{k-1})\log(H)}{(k-1)k}} + 4\sqrt{\frac{S_{k-1}(k - S_{k-1})\log(H)}{(k-1)km}} \tag{54}$$

$$\leq -\frac{1 + 2S_{k-1}m}{9(k-1)km} + \frac{6\log(H)}{m} + 4\sqrt{\frac{S_{k-1}(k - S_{k-1})\log(H)}{(k-1)km}}. \tag{55}$$

where we use $\sqrt{a + b} \leq \sqrt{a} + \sqrt{b}$ for any $a, b \geq 0$.

If $S_{k-1} = 0$, (55) implies that

$$\delta_k \leq -\frac{1}{9(k-1)km} + \frac{6\log(H)}{m} \leq \frac{1}{5k}. \tag{56}$$

where the final step is true as long as $m \geq 30K \log(H)$.

If $S_{k-1} = k - 1$, one has

$$\delta_k \leq -\frac{2}{9k} + \frac{6 \log(H)}{m} \leq -\frac{1}{5k}, \tag{57}$$

provided $m \gg K \log(H)$.

Otherwise, we obtain

$$\delta_k \leq -\frac{2}{9} \frac{S_{k-1}}{(k-1)k} + \frac{6 \log(H)}{m} + 4 \sqrt{\frac{S_{k-1}(k - S_{k-1}) \log(H)}{(k-1)km}} \leq -\frac{1}{9} \frac{S_{k-1}}{(k-1)k} < 0, \tag{58}$$

where the last step holds as long as $m \gg K^3 \log(H) \geq \log(H)k(k-1)(k - S_{k-1})/S_{k-1}$.

– Let us proceed to consider the case $k \in \mathcal{T}$.

First, the expected difference $\mathbb{E}[\delta_k]$ can be bounded by

$$
\begin{aligned}
\mathbb{E}[\delta_k] &= \frac{1}{m} \sum_{i=0}^{m-1} \frac{1 + 2S_{k-1}m + m + i}{2 + 2(k-1)m + m + i} - \frac{1}{m} \sum_{i=0}^{m-1} \frac{1 + 2S_{k-1}m + i}{2 + 2(k-1)m + i} \\
&= \sum_{i=0}^{m-1} \frac{1 + 2(k-1-S_{k-1})m}{\big(2 + 2(k-1)m + i\big)\big(2 + 2(k-1)m + m + i\big)} \\
&\geq \int_0^m \frac{1 + 2(k-1-S_{k-1})m}{\big(2 + 2(k-1)m + x\big)\big(2 + 2(k-1)m + m + x\big)} \, \mathrm{d}x \\
&= \frac{1}{m}\big(1 + 2(k-1-S_{k-1})m\big) \log\left(1 + \frac{m^2}{4\big(1 + (k-1)m\big)(1 + km)}\right) \\
&\geq \frac{1 + 2(k-1-S_{k-1})m}{9(k-1)km}, \tag{59}
\end{aligned}
$$

where the last step holds as $m \gg 1$ and $\log(1 + x) \geq x/2$ for any $x \in [0,1]$.

Next, it is straightforward to compute the variance

$$
\begin{aligned}
V_k &:= \sum_{i=1}^{2m} \mathrm{Var}\big(\chi_{2(k-1)m+i}\big) = \sum_{i=1}^{m} t_{2(k-1)m+i}(1 - t_{2(k-1)m+i}) \\
&= t_{2(k-1)m+1}\big(1 - t_{2(k-1)m+1}\big) + \big(1 + 2(k-1-S_{k-1})m\big) \sum_{i=1}^{2m-1} \frac{1 + 2S_{k-1}m + i}{\big(2 + 2(k-1)m + i\big)^2}.
\end{aligned}
$$

Applying the same argument for (53), we can bound

$$
\begin{aligned}
V_k &\leq \frac{1}{4} + \big(1 + 2(k-1-S_{k-1})m\big) \int_0^{2m} \frac{1 + 2S_{k-1}m + x}{\big(2 + 2(k-1)m + x\big)^2} \, \mathrm{d}x \\
&\leq \frac{1}{4} + \frac{m\big(1 + 2(k-1-S_{k-1})m\big)\big(1 + 2(S_{k-1}+1)m\big)}{2\big(1 + (k-1)m\big)(1 + km)} \\
&\leq \frac{1}{4} + \frac{\big(1 + 2(k-1-S_{k-1})m\big)(S_{k-1}+1)}{(k-1)k}. \tag{60}
\end{aligned}
$$

Combining (59) and (60) with the Bernstein inequality yields that with probability at least $1 - H^{-2}$,

$$
\begin{aligned}
\delta_k &\geq \mathbb{E}[\delta_k] - \frac{4}{3}\frac{\log(H)}{m} - \frac{2}{m}\sqrt{V_k \log(H)} \\
&\geq \frac{1 + 2(k-1-S_{k-1})m}{9(k-1)km} - \frac{4}{3}\frac{\log(H)}{m} \\
&\quad - \frac{2}{m}\sqrt{\frac{\log(H)}{4} + \frac{\big(1 + 2(k-1-S_{k-1})m\big)(S_{k-1}+1)\log(H)}{(k-1)k}} \\
&\geq \frac{1 + 2(k-1-S_{k-1})m}{9(k-1)km} - \frac{7}{3}\frac{\log(H)}{m} - \frac{2\sqrt{2}}{m}\sqrt{\frac{(S_{k-1}+1)\log(H)}{(k-1)k}} \\
&\quad + 4\sqrt{\frac{(k-1-S_{k-1})(S_{k-1}+1)\log(H)}{(k-1)km}} \\
&\geq \frac{1 + 2(k-1-S_{k-1})m}{9(k-1)km} - \frac{6\log(H)}{m} - 4\sqrt{\frac{(k-1-S_{k-1})(S_{k-1}+1)\log(H)}{(k-1)km}}.
\end{aligned}
$$

(61)

(62)

(63)

where we use $\sqrt{a+b} \leq \sqrt{a} + \sqrt{b}$ for any $a, b \geq 0$.

If $S_{k-1} = 0$, we know from (55) that

$$
\delta_k \geq \frac{2}{9k} - \frac{6\log(H)}{m} - 4\sqrt{\frac{\log(H)}{km}} > \frac{1}{5k},
$$

(64)

where the last step follows from $m \gg K\log(H)$.

If $S_{k-1} = k-1$, one knows that

$$
\delta_k \geq \frac{1}{9(k-1)km} - \frac{6\log(H)}{m} \geq -\frac{1}{5k}.
$$

(65)

where the final inequality is true as long as $m \geq 30K\log(H)$.

Otherwise, we obtain

$$
\begin{aligned}
\delta_k &\geq \frac{2}{9}\frac{k-1-S_{k-1}}{(k-1)k} - \frac{6\log(H)}{m} - 4\sqrt{\frac{(k-1-S_{k-1})(S_{k-1}+1)\log(H)}{(k-1)km}} \\
&\geq \frac{1}{9}\frac{k-1-S_{k-1}}{(k-1)k} > 0,
\end{aligned}
$$

(66)

where the last step holds as long as $m \gg K^3\log(H) \geq \big(k(k-1)(k-S_{k-1})/S_{k-1}\big)\log(H)$.

 – When $S_{k-1} = 0$, combining (56) and (64) shows that adding arm $k$ to $\mathcal{T}$ if $\delta_k > 1/(5k)$ correctly identifies whether $k \in \mathcal{T}$ with probability at least $1 - H^{-2}$.
When $S_{k-1} = k-1$, putting (57) and (65) together reveals that adding arm $k$ to $\mathcal{T}$ if $\delta_k > -1/(5k)$ correctly tests whether $k \in \mathcal{T}$ with probability exceeding $1 - H^{-2}$.
Otherwise, collecting (58) and (66) together demonstrates that adding arm $k$ to $\mathcal{T}$ if $\delta_k > 0$ correctly determines whether $k \in \mathcal{T}$ with probability at least $1 - H^{-2}$.

Finally, we can use an induction argument to conclude with probability exceeding $1 - KH^{-2}$, all arms outside the trust set have been eliminated after the first stage. In other words, by defining the event

$$
\mathcal{E}_{\mathsf{ts}} := \big\{\widehat{\mathcal{T}} = \mathcal{T}\big\},
$$

(67)

we have

$$
\mathbb{P}\{\mathcal{E}_{\mathsf{ts}}^{\mathsf{c}}\} \leq 2KH^{-2}.
$$

(68)

In what follows, we shall work on the event $\mathcal{E}_{\mathsf{ts}}$.

**Stage 2: trust-aware exploration-exploitation.** By Stage 1 of Algorithm 1, the trust level at time $H_0 + 1$ satisfies $t_{H_0+1} = S_K/K = |\mathcal{T}|/K$, where we recall $S_K := \sum_{k=1}^K \mathbb{1}\{k \in \mathcal{T}\}$ counts the number of the arms that belong to the trust set $\mathcal{T}$. Similar to (41), we can express

$$1 - t_h = \left(1 - \frac{1}{1+h}\right)(1 - t_{h-1}) + \frac{1}{1+h}\mathbb{1}\{a_{h-1}^{\mathsf{pm}} \notin \mathcal{T}\} = \left(1 - \frac{1}{1+h}\right)(1 - t_{h-1}), \forall h > H_0 + 1.$$

Here, the last line is true under the event $\mathcal{E}_{\mathsf{ts}}$. Therefore, one can then use induction to obtain that for each $h > H_0$,

$$1 - t_h = \frac{H_0 + 2}{h + 1}(1 - t_{H_0+1}) = \frac{H_0 + 2}{h + 1}\frac{K - S_K}{K}. \tag{69}$$

In particular, $t_h$ is an increasing function in $h$ when $h > H_0$.

With this in place, we are ready to control the regret. By (14) and (16), one can derive

$$\mathsf{reg}_H = \underbrace{\sum_{h=1}^{H_0}\left(r^\star - r(a_h^{\mathsf{ac}})\right)}_{\mathsf{reg}_H^{\mathsf{ts}}} + \underbrace{\sum_{h=H_0+1}^{H} t_h\left(r^\star - r(a_h^{\mathsf{pm}})\right)}_{\mathsf{reg}_H^{\mathsf{ta\text{-}pm}}} + \underbrace{\sum_{h=H_0+1}^{H}(1 - t_h)(r^\star - r_h^{\mathsf{own}})}_{\mathsf{reg}_H^{\mathsf{ta\text{-}own}}}.$$

In what follows, we shall control $\mathsf{reg}_H^{\mathsf{ts}}$, $\mathsf{reg}_H^{\mathsf{ta\text{-}pm}}$ and $\mathsf{reg}_H^{\mathsf{ta\text{-}own}}$ separately.

- Let us start with $\mathsf{reg}_H^{\mathsf{ts}}$. By our choice of the round length $m$, it is easy to bound

$$\mathsf{reg}_H^{\mathsf{ts}} \le H_0 = 2mK = 2K^4 \log(H). \tag{70}$$

- To bound $\mathsf{reg}_H^{\mathsf{pm}}$, let us first recall the notation $\Delta(a) := r^\star - r(a)$ for any $a \in [K]$. In addition, we define

$$N_s^{\mathsf{pm}}(a) := 1 \vee \sum_{i=H_0+1}^{s-1} \mathbb{1}\{a_i^{\mathsf{pm}} = a\}, \tag{71}$$

for any $a \in [K]$ and $s > H_0$. With these notations in hand, it is straightforward to derive

$$\mathsf{reg}_H^{\mathsf{ta\text{-}pm}} = \sum_{a \in \widehat{\mathcal{T}}} \Delta(a) \sum_{h=H_0+1}^{H} t_h \mathbb{1}\{a_h^{\mathsf{pm}} = a\}. \tag{72}$$

This suggests we need to control $\sum_{h=H_0+1}^{H} t_h \mathbb{1}\{a_h^{\mathsf{pm}} = a\}$. Towards this, let us fix an $a \in [K]$ such that $r(a) < r^\star$. Recall the definition of $\chi_h$, which obeys

$$\chi_h \mid \mathcal{F}_{h-1} \sim \mathsf{Bern}(t_h).$$

Let us define a sequence of random variables $X_h := \chi_h \mathbb{1}\{a_h^{\mathsf{pm}} = a\} - t_h \mathbb{1}\{a_h^{\mathsf{pm}} = a\}$, $h \ge 1$. Straightforward calculation yields that

$$\mathbb{E}\left[X_h \mid \mathcal{F}_{h-1}\right] = 0.$$

The sum of the conditional variances can be controlled by

$$\sum_{h=H_0+1}^{H} \mathbb{E}\left[X_h^2 \mid \mathcal{F}_{h-1}\right] = \sum_{h=H_0+1}^{H} \mathbb{1}\{a_h^{\mathsf{pm}} = a\}t_h(1 - t_h) \le \sum_{h=H_0+1}^{H}(1 - t_h)$$

$$= \sum_{h=H_0+1}^{H} \frac{H_0 + 2}{h + 1}\frac{K - S_K}{K}$$

$$\le \int_{H_0}^{H}(H_0 + 2)\frac{K - S_K}{K}\frac{1}{x + 1}\,\mathrm{d}x$$

$$\le (H_0 + 2)\frac{K - S_K}{K}\log\left(\frac{H + 1}{H_0 + 1}\right)$$

$$\le 2H_0 \log(H) \le c_1 K^4 \log^2(H),$$

where we use (69) under the event $\mathcal{E}_{\text{ts}}$ in the second line, and $c_1 > 0$ is some universal constant. Applying Lemma 1 by taking $\delta = H^{-1}$, $R = 1$, and $\sigma = c_1 K^4 \log^2(H)$, we find that with probability $1 - H^{-1}$,

$$\sum_{h=H_0+1}^{H} \chi_h \mathbb{1}\{a_h^{\text{pm}} = a\} \geq \sum_{h=H_0+1}^{H} t_h \mathbb{1}\{a_h^{\text{pm}} = a\} - \sqrt{3c_1} K^2 \log(H). \qquad (73)$$

Moreover, let us define the event

$$\mathcal{E}_{\text{ta-pm}} := \left\{ \left| \sum_{i=H_0+1}^{h} \left(R_i(a) - r(a)\right) \mathbb{1}\{a_i^{\text{ac}} = a\} \right| \leq \sqrt{N_{h+1}^{\text{ac}}(a) \log(H)}, \ \forall a \in [K], h > H_0 \right\}. \qquad (74)$$

We claim that under the event $\mathcal{E}_{\text{ta-pm}}$, the following holds for all $a \in [K]$ such that $r(a) < r^\star$:

$$\sum_{h=H_0+1}^{H} \chi_h \mathbb{1}\{a_h^{\text{pm}} = a\} \leq \left\lceil \frac{16 \log(H)}{\Delta^2(a)} \right\rceil. \qquad (75)$$

To see this, suppose that $\sum_{h=H_0+1}^{H} \chi_h \mathbb{1}\{a_h^{\text{pm}} = a\} > \left\lceil \frac{16 \log(H)}{\Delta^2(a)} \right\rceil$ for some $a \in [K]$ satisfying $r(a) < r^\star$. Then there exists an $H' \leq H$ such that

$$\sum_{h=H_0+1}^{H'} \chi_h \mathbb{1}\{a_h^{\text{pm}} = a\} = \left\lceil \frac{16 \log(H)}{\Delta^2(a)} \right\rceil.$$

Now, for any $h \geq H'$, one knows from the definition of $N_h^{\text{ac}}(a)$ in (5) that

$$N_{h+1}^{\text{ac}}(a) \geq N_{H'+1}^{\text{ac}}(a) = \sum_{h=H_0+1}^{H'} \chi_h \mathbb{1}\{a_h^{\text{ac}} = a\}$$

$$= \sum_{h=H_0+1}^{H'} \chi_h \mathbb{1}\{a_h^{\text{pm}} = a\} + \sum_{h=H_0+1}^{H'} (1 - \chi_h) \mathbb{1}\{a_h^{\text{own}} = a\} \geq \left\lceil \frac{16 \log(H)}{\Delta^2(a)} \right\rceil, \qquad (76)$$

where the second equality follows from the assumption on decision implementation deviations in (1).

On the other hand, note that the UCB estimator (6) is constructed based on $N_h^{\text{ac}}(a)$. Under the event $\mathcal{E}_{\text{ta-pm}}$, one can derive

$$\text{UCB}_h(a) = \frac{1}{N_{h+1}^{\text{ac}}(a)} \sum_{i=H_0+1}^{h} R_i(a) \mathbb{1}\{a_i^{\text{ac}} = a\}$$

$$\overset{(i)}{\leq} r(a) + \sqrt{\frac{\log(H)}{N_{h+1}^{\text{ac}}(a)}} \overset{(ii)}{\leq} r(a) + \frac{1}{4}\Delta(a)$$

$$< r(a^\star) - \frac{1}{4}\Delta(a) \overset{(iii)}{\leq} r(a^\star) - \sqrt{\frac{\log(H)}{N_{h+1}^{\text{ac}}(a^\star)}}$$

$$\overset{(iv)}{\leq} \frac{1}{N_{h+1}^{\text{ac}}(a^\star)} \sum_{i=H_0+1}^{h} R_i(a^\star) \mathbb{1}\{a_i^{\text{ac}} = a^\star\} = \text{UCB}_h(a^\star).$$

where (i) holds under the event (74); (ii) and (iii) are due to (76); (iv) arises from (74). By the arm selection criterion of the UCB algorithm, this implies that $a_h^{\text{pm}} \neq a$ for all $h > H'$. This further leads to

$$\sum_{h=H_0+1}^{H} \chi_h \mathbb{1}\{a_h^{\text{pm}} = a\} = \sum_{h=H_0+1}^{H'} \chi_h \mathbb{1}\{a_h^{\text{pm}} = a\} + \sum_{h=H'}^{H} \chi_h \mathbb{1}\{a_h^{\text{pm}} = a\} = \left\lceil \frac{16 \log(H)}{\Delta^2(a)} \right\rceil,$$

which contradicts the assumption. Therefore, this proves the claim (75).

In addition, recognize that $\left\{ \left( R_i(a) - r(a) \right) \mathbb{1}\{a_i^{\mathsf{ac}} = a\} \right\}_{i \geq 1}$ is a sequence of martingale differences with respect to the filtration $(\mathcal{F}_i)_{i \geq 0}$. It is straightforward to see that

$$\mathbb{E}\left[ \left( R_i(a) - r(a) \right) \mathbb{1}\{a_i^{\mathsf{ac}} = a\} \mid \mathcal{F}_{i-1} \right] = 0;$$

$$\sum_{i=1}^{h} \mathbb{E}\left[ \left( R_i(a) - r(a) \right)^2 \mathbb{1}\{a_i^{\mathsf{ac}} = a\} \mid \mathcal{F}_{i-1} \right] \leq \sum_{i=1}^{h} \mathbb{E}\left[ \mathbb{1}\{a_i^{\mathsf{ac}} = a\} \mid \mathcal{F}_{i-1} \right] = N_{h+1}^{\mathsf{ac}}(a).$$

We can then invoke the Azuma-Hoeffding inequality to obtain that for any fixed $h \geq H'$, with probability at least $1 - 2H^{-2}$,

$$\left| \sum_{i=H_0+1}^{h} \left( R_i(a) - r(a) \right) \mathbb{1}\{a_i^{\mathsf{ac}} = a\} \right| \leq \sqrt{N_{h+1}^{\mathsf{ac}}(a) \log(H)}.$$

Combined with the union bound, we find that

$$\mathbb{P}\{\mathcal{E}_{\mathsf{ta\text{-}pm}}^{\mathsf{c}}\} \leq 2KH^{-1}. \tag{77}$$

As a result, combining (73), (75), and (77) reveals that with probability at least $1 - O(KH^{-1})$, for all $a \in [K]$ such that $r(a) < r^\star$:

$$\sum_{h=H_0+1}^{H} t_h \mathbb{1}\{a_h^{\mathsf{pm}} = a\} \leq c_2 \left( \frac{\log(H)}{\Delta^2(a)} + K^2 \log(H) \right) \wedge H, \tag{78}$$

where $c_2 > 0$ is some numerical number. Plugging (78) back into (72) and taking $\Delta = \sqrt{K \log(H)/H}$, we obtain with probability exceeding $1 - O(KH^{-1})$,

$$\mathsf{reg}_H^{\mathsf{ta\text{-}pm}} = \sum_{a \in \mathcal{T}:\Delta(a) \leq \Delta} \Delta(a) \sum_{h=H_0+1}^{H} t_h \mathbb{1}\{a_h^{\mathsf{pm}} = a\} + \sum_{a \in \mathcal{T}:\Delta(a) > \Delta} \Delta(a) \sum_{h=H_0+1}^{H} t_h \mathbb{1}\{a_h^{\mathsf{pm}} = a\}$$

$$\leq \Delta H + \sum_{a \in \mathcal{T}:\Delta(a) > \Delta} \Delta(a) \sum_{h=H_0+1}^{H} t_h \mathbb{1}\{a_h^{\mathsf{pm}} = a\}$$

$$\leq \Delta H + c_2 \sum_{a \in \mathcal{T}:\Delta(a) > \Delta} \Delta(a) \left( \frac{\log(H)}{\Delta^2(a)} + K^2 \log(H) \right)$$

$$\leq \Delta H + c_2 \frac{|\mathcal{T}|}{\Delta} \log(H) + c_2 |\mathcal{T}| K^2 \log(H)$$

$$\leq 2\sqrt{c_2 |\mathcal{T}| H \log(H)} + c_2 |\mathcal{T}| K^2 \log(H). \tag{79}$$

- It remains to control $\mathsf{reg}_H^{\mathsf{ta\text{-}own}}$. By the trust bound in (69) under the event $\mathcal{E}_{\mathsf{ts}}$, we can show that

$$\mathsf{reg}_H^{\mathsf{ta\text{-}own}} = \sum_{h=H_0+1}^{H} (1 - t_h) \Delta_h^{\mathsf{own}} \overset{(i)}{\leq} \sum_{h=H_0+1}^{H} (1 - t_h)$$

$$\overset{(ii)}{\leq} \sum_{h=H_0+1}^{H} \frac{H_0 + 2}{h + 1} \frac{K - |\mathcal{T}|}{K}$$

$$\leq \int_{H_0}^{H} (H_0 + 2) \frac{K - |\mathcal{T}|}{K} \frac{1}{x + 1} \, \mathrm{d}x$$

$$\leq (H_0 + 2) \frac{K - |\mathcal{T}|}{K} \log\left( \frac{H + 1}{H_0 + 1} \right)$$

$$\leq 2H_0 \frac{K - |\mathcal{T}|}{K} \log(H) \leq c_1 K^3 (K - |\mathcal{T}|) \log^2(H). \tag{80}$$

Here, (i) follows from $\Delta_h^{\mathsf{own}} \leq 1$ for all $h \geq 1$; (ii) arises from (69); the last step follows from the choice of the round length $m$.

- Combining (70), (79), and (80) yields that with probability at least $1 - O(KH^{-1})$,

$$\mathsf{reg}_H \lesssim K^4 \log(H) + \sqrt{|\mathcal{T}|H\log(H)} + |\mathcal{T}|K^2\log(H) + K^3(K - |\mathcal{T}|)\log^2(H) \quad (81)$$

$$\lesssim \sqrt{KH\log(H)} + K^4\log^2(H).$$

**Combining Stage 1 and Stage 2.**  Finally, we can bound

$$\mathbb{E}[\mathsf{reg}_H] \lesssim \sqrt{KH\log(H)} + K^4\log^2(H) + KH^{-1}H$$

$$\leq C_1\sqrt{KH\log(H)} + C_2K^4\log^2(H)$$

for some constants $C_1, C_2$ independent of $H$ and $K$. This concludes the proof.

## D  PROOF OF THEOREM 3

Fix an arbitrary admissible policy $\pi^{\mathsf{pm}}$. We will construct two trust-aware $K$-armed bandit instances and use the superscripts $(1)$ and $(2)$ to distinguish the quantities associated with the first and second instances, respectively.

Let us denote by $\mathcal{D} := \left(a_h^{\mathsf{pm}}, a_h^{\mathsf{ac}}, a_h^{\mathsf{own}}, R_h(a_h^{\mathsf{ac}}), \chi_h\right)_{i\in[H]}$ and $\widetilde{\mathcal{D}} := \left(a_h^{\mathsf{pm}}, a_h^{\mathsf{ac}}, R_h(a_h^{\mathsf{ac}}), \chi_h\right)_{i\in[H]}$. Let $\mathbb{P}^{(1)}$ and $\widetilde{\mathbb{P}}^{(1)}$ denote the probability distribution of the random variables in $\widetilde{\mathcal{D}}$ and $\mathcal{D}$ in the first instance, respectively. The probability distributions $\mathbb{P}^{(2)}$ and $\widetilde{\mathbb{P}}^{(2)}$ are defined similarly for the second instance. Also, we use $\mathbb{E}^{(1)}$ and $\mathbb{E}^{(2)}$ to denote the associated expectations.

For the MABs, the random rewards are chosen to be Bernoulli random variables where $R_h(k) \sim \mathsf{Bern}\left(r(k)\right)$ for all $k \in [K]$ and $h \geq 1$. For the first instance, we let $r^{(1)}(1) = 1/2 + \Delta$ and $r^{(1)}(k) = 1/2$ for any $k \neq 1$, where $0 < \Delta \leq 1/8$ will be specified later. Moreover, let $k_0 \neq 1$ be some arm such that $\mathbb{E}^{(1)}[N_{H+1}^{\mathsf{ac}}(k)] \leq H/(K-1)$ under the policy $\pi^{\mathsf{pm}}$. As for the second instance, we let $r^{(2)}(1) = 1/2 + \Delta$, $r^{(2)}(k_0) = 1/2 + 2\Delta$, and $r^{(2)}(k) = 1/2$ otherwise. Finally, the trust set $\mathcal{T}$ and own policy $\pi^{\mathsf{own}}$ are chosen to be the same in the two instances.

With these definitions in hand, we can express the probability density $p^{(1)}$ of $\mathbb{P}^{(1)}$ as

$$p^{(1)}(\mathcal{D}) = \prod_{i=1}^{H} p^{(1)}(a_h^{\mathsf{pm}} \mid \mathcal{F}_{i-1}) p^{(1)}(a_h^{\mathsf{own}} \mid \mathcal{F}_{i-1}) p^{(1)}(\chi_h \mid \mathcal{F}_{i-1}) p^{(1)}(a_h^{\mathsf{ac}} \mid a_h^{\mathsf{pm}}, a_h^{\mathsf{own}}, \chi_h) p^{(1)}\left(R_h(a_h^{\mathsf{ac}}) \mid a_h^{\mathsf{ac}}\right).$$

Since $\pi^{\mathsf{pm}}$ is fixed and $\pi^{\mathsf{own}}$ and $\mathcal{T}$ are the same in the two instances, we have

$$\log \frac{d\mathbb{P}^{(1)}}{d\mathbb{P}^{(2)}}(\mathcal{D}) = \sum_{i=1}^{H} \log \frac{p^{(1)}\left(R_h(a_h^{\mathsf{ac}}) \mid a_h^{\mathsf{ac}}\right)}{p^{(2)}\left(R_h(a_h^{\mathsf{ac}}) \mid a_h^{\mathsf{ac}}\right)}.$$

Taking the expectation with respect to $\mathbb{P}^{(1)}$ yields

$$\mathsf{KL}\left(\mathbb{P}^{(1)} \parallel \mathbb{P}^{(2)}\right) = \mathbb{E}^{(1)}\left[\sum_{i=1}^{H} \log \frac{p^{(1)}\left(R_h(a_h^{\mathsf{ac}}) \mid a_h^{\mathsf{ac}}\right)}{p^{(2)}\left(R_h(a_h^{\mathsf{ac}}) \mid a_h^{\mathsf{ac}}\right)}\right]$$

$$\overset{(i)}{=} \sum_{k=1}^{K} \mathbb{E}^{(1)}[N_{H+1}^{\mathsf{ac}}(k)] \mathsf{KL}\left(\mathsf{Bern}\left(r^{(1)}(k)\right) \parallel \mathsf{Bern}\left(r^{(2)}(k)\right)\right)$$

$$\overset{(ii)}{=} \mathbb{E}^{(1)}[N_{H+1}^{\mathsf{ac}}(k_0)] \mathsf{KL}\left(\mathsf{Bern}\left(r^{(1)}(k_0)\right) \parallel \mathsf{Bern}\left(r^{(2)}(k_0)\right)\right)$$

$$\overset{(iii)}{\leq} \frac{H}{K-1} \mathsf{KL}\left(\mathsf{Bern}(1/2) \parallel \mathsf{Bern}(1/2 + 2\Delta)\right)$$

$$\overset{(iv)}{\lesssim} \frac{H}{K-1}\Delta^2.$$

Here, (i) follows from the divergence decomposition in Lattimore & Szepesvári (2020, Lemma 15.1); (ii) and (iii) are due to the construction of the instances; (iv) follows from Lemma 2 below and $\Delta \leq 1/8$.

**Lemma 2.** *For any $a, b \in [0, 1]$, let $\mathsf{Bern}(a)$ and $\mathsf{Bern}(b)$ denote two Bernoulli distributions with parameters $a$ and $b$, respectively. Then one has*

$$\mathsf{KL}(\mathsf{Bern}(a) \parallel \mathsf{Bern}(b)) \leq \frac{(a-b)^2}{b(1-b)}. \tag{82}$$

*In addition, if $|b - 1/2| \leq 1/4$, then one further has*

$$\mathsf{KL}(\mathsf{Bern}(a) \parallel \mathsf{Bern}(b)) \leq 8(a-b)^2. \tag{83}$$

Consequently, by choosing $\Delta \asymp \sqrt{K/H}$, we obtain

$$\mathsf{KL}(\mathbb{P}^{(1)} \parallel \mathbb{P}^{(2)}) \leq \frac{H\Delta^2}{K-1} \lesssim 1. \tag{84}$$

Moreover, from the data processing inequality (Cover, 1999), we can further control

$$\mathsf{KL}(\widetilde{\mathbb{P}}^{(1)} \parallel \widetilde{\mathbb{P}}^{(2)}) \leq \mathsf{KL}(\mathbb{P}^{(1)} \parallel \mathbb{P}^{(2)}) \lesssim 1. \tag{85}$$

Therefore, applying the standard reduction scheme (see e.g., Tsybakov (2008)), we conclude that

$$\inf_{\pi} \sup_{\Pi(K, \mathcal{T}, \pi^{\mathsf{own}})} \mathbb{E}[\mathsf{reg}_H(\pi)] \gtrsim H\Delta \exp\left(-\mathsf{KL}(\widetilde{\mathbb{P}}^{(1)} \parallel \widetilde{\mathbb{P}}^{(2)})\right) \gtrsim \sqrt{KH}.$$

## E    EXTENSION TO GENERAL TRUST SET

We now briefly discuss the scenario where Assumption 1 does not hold, i.e., where no optimal arm is included in the trust set. This extension accommodates the practical applications where implementer limitations may preclude optimal arms from the trust set.

First, we present Theorem 4 below, which characterizes the minimax lower bound of the regret in this setting.

**Theorem 4.** *Let $\varepsilon := r^\star - \max_{k \in \mathcal{T}} r(k)$. For any algorithm that outputs a policy $\pi$, there exists a MAB instance, a trust set, and an own policy such that*

$$\mathbb{E}[\mathsf{reg}_H(\pi)] \geq c\varepsilon H, \tag{86}$$

*for some absolute constant $c > 0$. In particular, for any $K \geq 2$, one has*

$$\inf_{\pi} \sup_{\Pi(K, \mathcal{T}, \pi^{\mathsf{own}})} \mathbb{E}[\mathsf{reg}_H(\pi)] \geq c(\sqrt{KH} \vee \varepsilon H). \tag{87}$$

In other words, if the best arm in the trust set is suboptimal by a constant gap, Theorem 4 demonstrates that all algorithms must incur a linear regret in the worst case.

Next, we present the theoretical guarantees of Algorithm 1 without Assumption 1.

**Theorem 5.** *For any $K \geq 2$, the expected regret of the policy $\pi$ generated by Algorithm 1 satisfies*

$$\sup_{\Pi(K, \mathcal{T}, \pi^{\mathsf{own}})} \mathbb{E}[\mathsf{reg}_H(\pi)] \leq C_1 \sqrt{KH \log(H)} + C_2 K^4 \log^2(H) + \varepsilon H, \tag{88}$$

*for some positive constants $C_1$ and $C_2$ independent of $H$ and $K$, where $\varepsilon := r^\star - \max_{k \in \mathcal{T}} r(k)$.*

In short, the proposed algorithm continues to attain the minimax regret (up to some logarithmic factors) when $H$ is sufficiently large. The additional term $\varepsilon H$ can be treated as the cost paid for the trust set not containing any optimal arm.

**Proof of Theorem 4.**    Fix an arbitrary policy $\pi$ and $\varepsilon > 0$. Let us construct a trust-aware MAB instance as follows. We choose $K = 2$, $r(1) = 1/2 - \varepsilon$, $r(2) = 1/2$, $R_h(k) = \mathsf{Bern}(r(k))$ for $k = 1, 2$, and $\pi_h^{\mathsf{own}}(1) = 1$ for all $h \geq 1$.

By the decomposition in (11), we know that

$$\mathbb{E}[\mathsf{reg}_H(\pi)] = \mathbb{E}\left[\sum_{h=1}^H t_h \varepsilon \mathbb{1}\{a_h^{\mathsf{pm}} = 1\}\right] + \mathbb{E}\left[\sum_{h=1}^H (1-t_h)\varepsilon\right] = \varepsilon H - \varepsilon \mathbb{E}\left[\sum_{h=1}^H t_h \mathbb{1}\{a_h^{\mathsf{pm}} = 2\}\right].$$

Suppose $N_{H+1}^{\mathsf{pm}}(2) \leq H/2$. Then we can use $t_h \leq 1$ to bound

$$\sum_{h=1}^{H} t_h \mathbb{1}\{a_h^{\mathsf{pm}} = 2\} \leq \sum_{h=1}^{H} \mathbb{1}\{a_h^{\mathsf{pm}} = 2\} = N_{H+1}^{\mathsf{pm}}(2) \leq \frac{H}{2}.$$

Alternatively, if $N_{H+1}^{\mathsf{pm}}(2) > H/2$ (or equivalently, $N_{H+1}^{\mathsf{pm}}(1) \leq H/2$), by the trust update rule

$$t_h = \frac{1 + N_h^{\mathsf{pm}}(1)}{h + 1},$$

we can bound

$$
\begin{aligned}
\sum_{h=1}^{H} t_h \mathbb{1}\{a_h^{\mathsf{pm}} = 2\} &= \sum_{h=1}^{H} \frac{1 + N_h^{\mathsf{pm}}(1)}{h + 1} \mathbb{1}\{a_h^{\mathsf{pm}} = 2\} \\
&\leq \sum_{h=1}^{3H/4} 1 + \sum_{h=3H/4+1}^{H} \frac{1 + N_{H+1}^{\mathsf{pm}}(1)}{1 + h} \\
&\leq \frac{3}{4} H + \frac{1}{4} H \frac{1 + H/2}{1 + 3H/4} \\
&\leq \frac{27}{28} H.
\end{aligned}
$$

Combining these two bounds, we obtain

$$\mathbb{E}[\mathsf{reg}_H(\pi)] \geq \frac{\varepsilon}{28} H.$$

**Proof of Theorem 5.** Let us denote $r^{\mathsf{ts}} := \max_{k \in \mathcal{T}} r(k)$, namely, the highest expected reward of the arms belonging to the trust set. We can then express

$$\mathsf{reg}_H = \sum_{h=1}^{H} \left(r^\star - r(a_h^{\mathsf{ac}})\right) = \sum_{h=1}^{H} (r^\star - r^{\mathsf{ts}}) + \sum_{h=1}^{H} \left(r^{\mathsf{ts}} - r(a_h^{\mathsf{ac}})\right). \tag{89}$$

By slightly modifying the proof analysis for (81), it can be shown that with probability at least $1 - O(KH^{-1})$,

$$\sum_{h=1}^{H} \left(r^{\mathsf{ts}} - r(a_h^{\mathsf{ac}})\right) \lesssim K^4 \log(H) + \sqrt{|\mathcal{T}| H \log(H)} + |\mathcal{T}| K^2 \log(H) + K^3 (K - |\mathcal{T}|) \log^2(H).$$

Combining this with $\sum_{h=1}^{H} (r^\star - r^{\mathsf{ts}}) \leq \varepsilon H$, Theorem 5 follows as an immediate consequence.

