# OpenReview forum: "Minimax-optimal trust-aware multi-armed bandits"
_ICLR.cc/2025/Conference — Submitted to ICLR 2025_

### Official Review · Reviewer_fPsh · 2024-11-03

**Soundness:** 3
**Presentation:** 3
**Contribution:** 3
**Rating:** 8
**Confidence:** 3

**Summary:**

This paper integrates human trust dynamics into the multi-armed bandit (MAB) framework to address the influence of trust on sequential decision-making. Recognizing that a lack of trust can cause users to deviate from recommended policies, potentially leading to suboptimal outcomes, the authors critique existing approaches for focusing primarily on learning in uncertain environments without accounting for human willingness. To address this gap, they propose a two-stage trust-aware MAB algorithm that adapts to the decision-maker’s trust level, aiming to build and maintain trust while efficiently identifying optimal actions. The authors provide a minimax regret bound for this trust-aware MAB model and highlight that common algorithms, such as Upper Confidence Bound (UCB), can incur near-linear regret when trust is compromised. Theoretical results show that the proposed algorithm achieves a regret bound of order $\sqrt{KH \log H}$ without requiring prior knowledge of the decision-maker’s trust level or policy.

**Strengths:**

1. This paper frames a new and interesting problem setting in MAB, which considers the decision-making problem with humans in the loop and accounts for more robust solutions when people do not trust the recommended actions.
2. To tackle this new problem, this paper proposes a novel UCB-based algorithm to solve the issue of lack of trust and provides rigorous theoretical analysis for the lower bound and the upper bound, as well as why the classical UCB algorithm fails.
3. To the best of my knowledge, this paper is the first work to develop a trust-aware algorithm that provably achieves near-minimax optimal statistical efficiency. The theoretical results are validated by empirical studies.
4. The presentation of the paper is clear and easy to follow.

**Weaknesses:**

I do not have major concerns about the novelty and contributions of the paper. However, some assumptions could be justified/relaxed to enhance the practical value of the work

1. From my perspective, Assumption 1 is quite strong in the sense that the implementor’s trust set $\mathcal{T}$ must contain one optimal arm with reward $r*$. In the recommender system, for example, there are thousands of products to recommend and it is not very realistic that the implementer can have access to such  $\mathcal{T}$. Can this assumption be relaxed if $\mathcal{T}$ only contains the optimal arm with some probability or suboptimal arm whose reward is bounded by $\epsilon$. This is more realistic since we may have some offline data before the learning game which can output the optimal arm with a certain probability or an arm with bounded suboptimal gap.
2. The current algorithm assumes that whether the user adopts the recommended policy $\mathcal{X}_h$ is a Bernoulli random variable and can be observed. However, in real applications, such feedback is not directly accessible as the implementor may not willing to reveal whether they take the recommended policy or not (suppose he does not trust others).  As such, I think the current algorithm may not work without this feedback.

**Questions:**

1. Can the current algorithm handle the case when the implementer's trust set only contains the optimal arm with some probability or the suboptimal arm whose suboptimality gap is bounded?
2. Can the authors give some practical applications where the implementer would like to reveal their choice on whether he/she follows the recommended policy? If the implementer chooses not to reveal this feedback, is there any way the learner can infer whether the implementer follows the recommended policy from other types of feedback?

---

> ### Author Response · Authors · 2024-11-16
>
> * Thank you for the insightful question.
> Assumption 1 allows us to focus on scenarios where sublinear regret is information-theoretically achievable. When the trust set does not contain any optimal arm, even perfect recommendations will inevitably reduce trust, incurring a linear regret in the worst case.
>
>     Following reviewers' suggestions, we added a section in the appendix to discuss the case where the trust set does not include any optimal arm.
>     Without Assumption 1, we show that the minimax lower bound becomes $\\Omega(\\sqrt{KH}+\\epsilon H)$ where $\\epsilon:=\\max_{k\\in\\mathcal T}r(k)-r^\\star$ is the suboptimality gap of the best arm in the trust set. This implies that when $\\epsilon$ is constant large, any algorithm must incur a linear regret in the worst case.
>
>     In addition, our algorithm remains robust in this setting. By slightly modifying our proof, we can show that the regret upper bound of the proposed algorithm becomes $\\widetilde{O}(\\sqrt{KH}+K^4+\\epsilon H)$. This demonstrates that our algorithm is still nearly minimax optimal (up to some logarithmic factors) when $H$ is sufficiently large. When the trust set contains moderately good arms ($\\epsilon=\\widetilde O(1/\\sqrt{H})$), it can still attain the classical minimax regret $\\widetilde{O}(\\sqrt{H})$. Similarly, when the optimal arm is included with some probability, the bounds above generalize by replacing $\\epsilon$  with the expected reward gap.
>
>
> * This is certainly another valid point.
>     - First, this assumption is motivated by various real-world human-autonomy interaction applications. One example arises from shared control in semi-autonomous driving [1]. While both the autonomous system and humans can execute the same actions (e.g., turning the steering wheel left), humans need to touch the wheel if they decide to take actions on their own. In this case, pressure sensors on the steering wheel can directly record whether humans follow or override autonomy, where human control corresponds to $\\chi_h=1$. Another example is human-robot collaboration to complete certain tasks [2,3], where recommendation compliance is indeed observed and recorded due to the collaborative nature of the missions. Hence, we adopt this observability assumption in our framework. As our work represents an initial exploration into trust-aware MABs, we believe this setup maintains practical relevance and implications.
>     - Next, we briefly discuss how to extend our algorithm where such observability is infeasible in certain scenarios. Recall that $\\chi_h$ is used in the first stage to estimate unknown trust $t_h$ because $t_h = \\mathbb E[\\chi_h]$. When $\\chi_h$ is unobservable, we can approximate it by $\\tilde\\chi_h\\coloneqq{1}\\{a^{pm}_h=a^{ac}_h\\}.$ As the reviewer pointed out, $\\tilde \\chi_h$ is no longer an unbiased estimator of $t_h$: given the recommended arm $a^{pm}$, we have $\\mathbb E[\\tilde\\chi_h]=t_h+\\pi^{own}_h(a^{pm})$.
> When $\\pi^{own}_h$ remains relatively stable over short periods, we can incorporate an additional procedure in the first stage of our algorithm to obtain a reasonable estimate of $\\pi^{own}_h(\\cdot)$. For example, the empirical distribution of the samples satisfying $a^{pm}_h \\neq a^{ac}_h$ can be used.
> Similar to the current setting, we can show that choosing the length of the first stage polynomial in $K$ would achieve the desired estimation accuracy, and only introduce an extra term (polynomial large in $K$) to the regret upper bound. Hence the regret of the resulting algorithm remains near-optimal with respect to $H$.
>     - Finally, we acknowledge that fully addressing the unobservable case requires additional work. As this paper serves as an initial attempt to build theoretical foundations for trust-aware decision-making, we choose to focus on the observable case---a condition that holds in many practical applications while conveying the core insights of this work. A comprehensive treatment of the unobservable scenario is an important direction for future work.
>
> [1] Wang, Chengshi, et al. "Haptic assistive control with learning-based driver intent recognition for semi-autonomous vehicles." IEEE Transactions on Intelligent Vehicles 8.1 (2021): 425-437.
>
> [2] Chen, Min, et al. "Planning with trust for human-robot collaboration." Proceedings of the 2018 ACM/IEEE international conference on human-robot interaction. 2018.
>
> [3] Bhat, Shreyas, et al. "Clustering trust dynamics in a human-robot sequential decision-making task." IEEE Robotics and Automation Letters 7.4 (2022): 8815-8822.

---

> > ### Comment · Reviewer_fPsh · 2024-11-16
> >
> > Thank you for the detailed and thoughtful response. The authors have adequately addressed my concerns by providing clear examples and revising the paper to relax the previously strong assumption. As a result, I no longer have further questions and have raised my score to 8. The paper would be even more impactful if the observability problem could also be addressed.

---

### Official Review · Reviewer_HakX · 2024-11-03

**Soundness:** 2
**Presentation:** 3
**Contribution:** 3
**Rating:** 6
**Confidence:** 3

**Summary:**

This paper formulates the problem of trust-aware stochastic bandits in which the algorithm cannot interact with the environment directly, but instead recommends an action to the implementer that then either plays the recommended action with probability $t_h$ or plays a different action with probability $1 - t_h$. The paper assumes that $t_h$ follows a dynamic model, where (roughly-speaking) $t_h$ increases if the algorithm plays an action in a (unknown) trusted set of actions, and decreases if the algorithm plays an action not in this set. They give an algorithm that first explores the actions to identify the set of trusted actions and then plays UCB. They show that this algorithm enjoys $\tilde{O}(\sqrt{KD} + K^4)$ regret and give a regret lower bound of $\Omega(\sqrt{KT})$. They also show that UCB suffers linear regret in this setting.

**Strengths:**

- The proposed algorithm provides a way to estimate the dynamical model with stochastic feedback in a small number of rounds. This appears to be a relatively difficult problem that is certainly more difficult than estimating the mean of a fixed distribution (as is typically done in explore-exploit problems for stochastic bandits).
- The paper shows (nearly)-minimax optimal regret.
- The presentation is good.

**Weaknesses:**

- I didn't see where the paper states the information available to the algorithm. It looks like the algorithm (in Algorithm 1) uses $\chi_h$. I don't think this is very realistic because the algorithm is observing the implementer's internal randomness. It would be more realistic if the algorithm observed the implementer's action $a_h^{ac}$. Note that this is not the same as observing $\chi_h$ because the implementer's policy could be the same as the algorithm's in some rounds and therefore $a_h^{pm} = a_h^{ac}$ even though $\chi_h$ might be $0$.
- I think the claims of $\tilde{O}(\sqrt{K T})$ are stated incorrectly, as (to my knowledge) it is standard to show all $K$ terms when one is shown, i.e. need to state it as $\tilde{O}(\sqrt{KT} + K^4)$.
- Although it is often stated as *nearly* minimax optimal regret, there are places that claim minimax optimal regret (e.g. the title) which I don't believe is a correct characterization given that the regret bounds are $\tilde{O}(\sqrt{KT} + K^4)$ and $\Omega (\sqrt{KT})$.

**Questions:**

- Can you clarify the reason for providing the algorithm with the implementer's internal random variable $\chi_h$?
- I suggest stating the regret upper bounds more precisely in the introduction.

---

> ### Author Response · Authors · 2024-11-16
>
> **Response to Weakness**
> * Yes, we assume the policy-maker has access to $(\\chi_h)_{h\\geq1}$.
>     - First, this assumption is motivated by various real-world human-autonomy interaction applications. One example arises from shared control in semi-autonomous driving [1]. While both the autonomous system and humans can execute the same actions (e.g., turning the steering wheel left), humans need to touch the wheel if they decide to take actions on their own. In this case, pressure sensors on the steering wheel can directly record whether humans follow or override autonomy, where human control corresponds to $\\chi_h=1$. Another example is human-robot collaboration to complete certain tasks [2,3], where recommendation compliance is indeed observed and recorded due to the collaborative nature of the missions. Hence, we adopt this observability assumption in our framework. As our work represents an initial exploration into trust-aware MABs, we believe this setup maintains practical relevance and implications.
>     - Next, we briefly discuss how to extend our algorithm where such observability is infeasible in certain scenarios. Recall that $\\chi_h$ is used in the first stage to estimate unknown trust $t_h$ because $t_h = \\mathbb E[\\chi_h]$. When $\\chi_h$ is unobservable, we can approximate it by $\\tilde\\chi_h\\coloneqq{1}\\{a^{pm}_h=a^{ac}_h\\}.$ As the reviewer pointed out, $\\tilde \\chi_h$ is no longer an unbiased estimator of $t_h$: given the recommended arm $a^{pm}$, we have $\\mathbb E[\\tilde\\chi_h]=t_h+\\pi^{own}_h(a^{pm})$.
> When $\\pi^{own}_h$ remains relatively stable over short periods, we can incorporate an additional procedure in the first stage of our algorithm to obtain a reasonable estimate of $\\pi^{own}_h(\\cdot)$. For example, the empirical distribution of the samples satisfying $a^{pm}_h \\neq a^{ac}_h$ can be used.
> Similar to the current setting, we can show that choosing the length of the first stage polynomial in $K$ would achieve the desired estimation accuracy, and only introduce an extra term (polynomial large in $K$) to the regret upper bound. Hence the regret of the resulting algorithm remains near-optimal with respect to $H$.
>     - Finally, we acknowledge that fully addressing the unobservable case requires additional work. As this paper serves as an initial attempt to build theoretical foundations for trust-aware decision-making, we choose to focus on the observable case---a condition that holds in many practical applications while conveying the core insights of this work. A comprehensive treatment of the unobservable scenario is an important direction for future work.
> * This is certainly a valid point. We have revised the statement of the regret bound to $\widetilde{O}(\sqrt{KH}+K^4)$.
> * Thanks for the suggestion. We have changed "minimax optimal" to "near-optimal" in the title and other places.
>
> [1] Wang, Chengshi, et al. "Haptic assistive control with learning-based driver intent recognition for semi-autonomous vehicles." IEEE Transactions on Intelligent Vehicles 8.1 (2021): 425-437.
>
> [2] Chen, Min, et al. "Planning with trust for human-robot collaboration." Proceedings of the 2018 ACM/IEEE international conference on human-robot interaction. 2018.
>
> [3] Bhat, Shreyas, et al. "Clustering trust dynamics in a human-robot sequential decision-making task." IEEE Robotics and Automation Letters 7.4 (2022): 8815-8822.

---

> > ### Author Response · Authors · 2024-11-25
> >
> > Dear reviewer,
> >
> > Thank you for your detailed review and valuable comments. We have carefully addressed your questions and would greatly appreciate it if you could confirm whether our responses have satisfactorily addressed your concerns or if any further clarification is needed.
> >
> > Thank you once again for your time and consideration!

---

> > ### Comment · Reviewer_HakX · 2024-11-25
> >
> > Thank you for your detailed response and revisions to the paper. Overall, my concerns have been addressed in terms of fixing the statement of the regret bound, and I think it's okay to leave the observability as future work. I will increasing my score to a 6 accordingly.
> >
> > However, part of my concern was that the information available to the policy-maker was not specified in the problem statement. I did not see this addressed in your response or the revisions.

---

> > > ### Author Response · Authors · 2024-11-25
> > >
> > > Thank you for your quick response and raised score. We will add the following remark in the updated version to explicitly specify the information available to the policy-maker: *"We assume the policy-maker has access to information on whether or not the implementer adopts the recommendations, i.e., $\{\chi_h\}_{h\geq 1}$, which is well motivated by numerous real-world applications in the human-robot interaction domain (e.g., [3])"*.

---

### Official Review · Reviewer_YixL · 2024-11-05

**Soundness:** 3
**Presentation:** 3
**Contribution:** 3
**Rating:** 8
**Confidence:** 4

**Summary:**

This paper investigates the trust-aware multi-armed bandit (MAB) problem by incorporating a dynamic trust model into the conventional MAB framework. The authors address the discrepancy that arises when humans may not fully adhere to recommended policies due to varying levels of trust. To mitigate the limitations of standard MAB algorithms in these trust-dependent scenarios, they propose a two-stage trust-aware UCB algorithm and theoretically derive minimax regret bounds. A simulation study illustrates the practical benefits of their approach.

**Strengths:**

1) The paper addresses a critical gap by incorporating human trust into MAB frameworks, which is relevant for real-world applications like human-robot interactions.

2) The authors establish the minimax lower bound for their trust-aware model and demonstrate that standard UCB algorithms can incur near-linear regret under trust-related deviations.

3) The authors develop a novel trust-aware UCB algorithm, supported by rigorous theoretical results, including a near-minimax optimal regret bound.

**Weaknesses:**

1) The trust model used in this paper (the disuse trust behavior model) is relatively simple, and its applicability to more complex trust frameworks may be limited.

2) While the regret bound of Algorithm 1 in Theorem 2 is nearly optimal with respect to the time horizon $H$, the dependence on $K$ could be improved. As the authors note in Remark 4, optimizing the trust set identification stage could help reduce the burn-in cost. Additionally, Assumption 1 might further improve the regret in the trust-aware exploration-exploitation stage: since in Stage 2, the algorithm only needs to explore the arms within the trust set, it’s possible that the regret could scale as $O(\sqrt{|\mathcal{T}| H \log H})$.

3) The numerical experiments, which consider $ K = 10 $ arms with a uniform own policy over two arms, seem limited. The authors may want to explore more complex own policies involving more arms. Furthermore, it is unclear why the trust-aware UCB (blue) outperforms the UCB without trust issue (green) in Figure 1.

**Questions:**

1) It would be great to include a formal proof demonstrating that, without Assumption 1 (possibly in combination with certain assumptions about the own policy), no algorithm can achieve sublinear regret.

1) Exploring specific types of implementers with defined own policies could be interesting; prior knowledge of these policies might further enhance the regret results.

---

> ### Author Response · Authors · 2024-11-16
>
> **Response to Weakness**
> * While our model does not cover all complex trust frameworks, we believe it serves as an initial step toward a theoretical understanding of trust-aware decision-making. It effectively shows that classical MAB algorithms may be inefficient when trust issues arise, highlights the necessity of trust-aware algorithms, and conveys our core principle of "building trust before learning" in algorithmic design. We leave the exploration of more complicated trust models to future work.
> * Thank you for the insightful observation. Our analysis actually guarantees that for a fixed trust set $\\mathcal{T}$, the proposed algorithm achieves a regret bound of $\\widetilde{O}(\\sqrt{|\\mathcal{T}|H}+K^4)$. The current bound in Theorem 2 results from taking the supremum over $|\\mathcal{T}|$ (which can equal $K$). In the updated version, we have modified the proof to clarify this point. As for the dependence on $K$, our future work will focus on refining the algorithm to achieve the optimal dependence, which may require developing a more sophisticated trust-building phase.
> * Due to space constraints, we only showed a representative example that best demonstrates how trust issues hurt vanilla UCB and highlights our method's effectiveness.
>     When the trust set consists of the best two arms, gaining trust becomes extremely difficult since the arms recommended in the early exploration stage are mostly suboptimal and do not belong to the trust set. As proved in Theorem 1, ignoring trust issues would lead to linear regrets in this case. Conversely, when the trust set contains many suboptimal arms, building trust is much easier.
>
>     In the appendix of the updated version, we included additional numerical experiments for the scenarios suggested, including (1) $\\pi^{own}_h(a^\\star)$ is increasing in $h$; (2) $\\mathcal{T}$ contains the best-50\% arms; (3) $\\mathcal{T}$ contains optimal and highly suboptimal arms; (4) $\\mathcal{T}$ includes only one of multiple optimal arms. As can be seen from the plots, our method achieves regret scaling as $\\sqrt{H}$, with comparable or superior performances to UCB algorithms.
>
>     Regarding trust-aware UCB's superior performance, this is due to the chosen own policy $\\pi^{own}$, which pulls the best two arms. Therefore, during the exploration stage before the optimal arm is identified, whenever the implementer executes $\\pi^{own}$, the regret incurred by $\\pi^{own}$ tends to be smaller than that of UCB without decision deviations. This explains why trust-aware UCB achieves a smaller regret, despite sharing the same order in $H$.
>
> **Response to Questions**
> * Thank you for the suggestion. Following reviewers' suggestions, we added a section in the appendix to discuss the case when the trust set does not include any optimal arm.
> In particular, we added a theorem to rigorously justify that without Assumption 1, the minimax lower bound becomes $\\Omega(\\sqrt{KH}+\\epsilon H)$ where $\\epsilon:=\\max_{k\\in\\mathcal T}r(k)-r^\\star$. This implies that when the best arm in the trust set has a constant large suboptimality gap, any algorithm must incur a linear regret in the worst case.
> * Thanks for the suggestion. Exploiting implementer behavior patterns is a promising avenue for improving the regret guarantees. For instance, it would be interesting to consider the following scenarios including (1) uniform random selection within the trust set, (2) only pulling arms outside the recommended set, and (3) the probability of choosing the optimal arm increasing over time. Incorporating this prior knowledge into the design of the trust set identification and trust-building stages could potentially yield more efficient algorithms. We leave the detailed theoretical investigation to future work.

---

### Official Review · Reviewer_DkLf · 2024-11-08

**Soundness:** 4
**Presentation:** 4
**Contribution:** 3
**Rating:** 8
**Confidence:** 4

**Summary:**

This paper considers a stochastic multi-armed bandit setting where the actions recommended by the bandit policy is not perfectly executed by the human actor. Specifically, they consider a setting with $K$ arms played over $H$ timesteps, where the human has some set $\mathcal{T} \subset [K]$ of arms that they trust. This trust set is unknown to the policymaker, leading to information asymmetry.

The likelihood that the human follows the recommendation made by the policymaker depends on the "trust" that the policymaker has built so far. The trust level $t_h$ is defined in terms of the percentage of the time that the policymaker has recommended an arm that belongs in the trust set $\mathcal{T}$.

The authors show that UCB can lead to supralinear regret above $H / \sqrt{\log (T)}$ for some combination of bandit, trust set, and human policy. Thus, the authors propose a two-stage "trust aware" UCB-based algorithm where the first stage involves uniform arm selection, to eliminate arms outside the trust set, then a second stage that performs exploration/exploitation similar to standard UCB. The authors show that the expected regret is on the order $O(\sqrt{KH})$ and run an experiment showing improved regret in a single setting with $K=10$ arms and a horizon length of $H=14,000$.

**Strengths:**

1. The authors consider an important problem, of imperfect execution by downstream actors during the implementation of a sequential decision-making algorithm. The model is reasonably formulated, with a changing "trust level" by the human depending on the policy recommendations so far. The notion of private information (i.e., the trust set) that is not visible to the policymaker is also reasonable and well-formulated.

2. The theoretical results are rigorous, first showing how the standard UCB algorithm fails in this setting, then showing $\sqrt{KH}$ minimax optimal regret for their proposed algorithm. Additionally it is useful to observe that when the time horizon $H$ increases, the larger polynomial regret $O(K^4 \log^2(H))$ for the trust-set identification stage is negligible; this is reasonable as often algorithms assume that $H$ is significantly larger than $K$.

3. The paper is well-written and generally well presented.

**Weaknesses:**

1. Empirical results are limited and fail to show robustness. There is only a single setting, where the number of arms is rather small ($K=10$), the trust set is extremely small (just 2 arms, $\mathcal{T} = \{9,10\}$), and the human has a very naive policy (uniform exploration, with no learning). The arms are also selected so that the trust set is the two highest-reward arms. How does performance compare when the trust set is 50\% of arms? What about a larger number of arms? What about when the trust set is the optimal arm and many very poor-performing arms?

Or what about the setting that you describe, where there may be multiple equally optimal arms, and the trust set only includes one of the optimal arms?

2. The assumption that the trust set includes the optimal arm is very strong, but I understand that is necessary for the analysis.

3. A more reasonable "trust level" would be based on performance, not just blindly whether or not the arms selected follow the humans' intuition of the a priori trust set. For example, this would include the case where there are multiple optimal arms $A,B$, the algorithm repeatedly recommends arm $A$, but only arm $B$ is in the humans' trust set --- the trust level should not monotonically go down in this case.

Another way of formulating this would be to have the algorithm designer be able to influence the trust set $\mathcal{T}$ to add more arms, which are shown to perform well.



Smaller comments

1. The plot is hard to read (especially in a printout); it would help to increase the font size.

**Questions:**

1. The trust level as formulated is very hard to improve. Would it make sense to add some decay, so that the trust level is only based on the past $L$ number of observations? (so that trust can be better improved over time)

---

> ### Author Response · Authors · 2024-11-16
>
> **Response to Weakness**
> * Due to space constraints, we only showed a representative example that best demonstrates how trust issues hurt vanilla UCB and highlights our method's effectiveness.
>     When the trust set consists of the best two arms, gaining trust becomes extremely difficult since the arms recommended in the early exploration stage are mostly suboptimal and do not belong to the trust set. As proved in Theorem 1, ignoring trust issues would lead to linear regrets in this case. Conversely, when the trust set contains many suboptimal arms, building trust is much easier.
>
>     In the appendix of the updated version, we included additional numerical experiments for the scenarios suggested, including (1) $\\pi^{own}_h(a^\\star)$ is increasing in $h$; (2) $\\mathcal{T}$ contains the best-50\% arms; (3) $\\mathcal{T}$ contains optimal and highly suboptimal arms; (4) $\\mathcal{T}$ includes only one of multiple optimal arms. As can be seen from the plots, our method achieves regret scaling as $\\sqrt{H}$, with comparable or superior performances to UCB algorithms.
>
> * Assumption 1 allows us to focus on scenarios where sublinear regret is information-theoretically achievable. When the trust set does not contain any optimal arm, even perfect recommendations will inevitably reduce trust, incurring a linear regret in the worst case.
>
>     Following reviewers' suggestions, we added a section in the appendix to discuss the case where the trust set does not include any optimal arm.
>     Without Assumption 1, we show that the minimax lower bound becomes $\\Omega(\\sqrt{KH}+\\epsilon H)$ where $\\epsilon:=\\max_{k\\in\\mathcal T}r(k)-r^\\star$ is the suboptimality gap of the best arm in the trust set. This implies that when $\\epsilon$ is constant large, any algorithm must incur a linear regret in the worst case.
>     In addition, our algorithm remains robust in this setting. By slightly modifying our proof, we can show that the regret upper bound of the proposed algorithm becomes $\\widetilde{O}(\\sqrt{KH}+K^4+\\epsilon H)$. This demonstrates that our algorithm is still nearly minimax optimal (up to some logarithmic factors) when $H$ is sufficiently large. When the trust set contains moderately good arms ($\\epsilon=\\widetilde O(1/\\sqrt{H})$), it can still attain the classical minimax regret $\\widetilde{O}(\\sqrt{H})$.
>
> * First, our intention is to cover as many decision deviation possibilities as possible so our model does not exclude the case where the implementer may strongly oppose certain high-reward arms. Nevertheless, even in such a case, as long as the trust set includes at least one optimal arm (B), our algorithm can still identify the trust set and recommend arm $B$. Since arms $A$ and $B$ have the same expected rewards, the algorithm achieves near-optimal regrets even though arm $A$ reduces trust and will not be pulled often.
>
>     Second, we fully agree other reasonable trust update rules are worth exploring, such as the performance-based rule where increasing trust when the empirical average of the recommended arm exceeds a certain threshold chosen by the implementer. However, we note that while specific algorithmic details may vary, our core principle of "building trust before learning" remains valid across different trust models. Due to space limits, we leave the detailed exploration of other trust update models for future works.
>
> * Thank you for the suggestion. We have increased the font size.
>
> **Response to Question**
>
> This is certainly an important extension. If the trust level is calculated over a sliding window of $L$ steps, the suboptimality of UCB remains. Due to its aggressive exploration, most recommended arms during initial exploration are suboptimal, driving trust near zero before the optimal arm is identified. Even with the $L$-step trust update rule, this near-zero trust issue remains, causing the implementer to follow their own policy and not pull those recommended suboptimal arms in practice. Hence, UCB keeps recommending those suboptimal arms and trust never increases again. On the other hand, our trust-aware algorithm will perform better in this setup. As trust levels increase faster, the $L$-step trust update accelerates both trust set identification and trust-building phases, leading to reduced regret.

---

> > ### Comment · Reviewer_DkLf · 2024-11-26
> >
> > Thank you for your detailed and helpful response. The additional experiments are helpful to show the generalizability of your approach in practice.
> >
> > I have increased my response from 6 to 8.

---

### Meta-Review · Area_Chair_w526 · 2024-12-20

**Metareview:**

This paper looks at some principal/agent problem where the principal recommends an action to play in some multi-armed bandit problem.

The agent follows the recommendation if its trust in the principal is high enough, and the trust increases if the recommendation belongs to some set of action and decreases otherwise.

The algorithm proposed is rather obvious. First identify the trust set by recommending several times the same action (and see if the trust increased or decreased). Then compute a MAB optimal strategy on the set of trusted arms.

I am way less thrilled by this paper than the reviewers, hence I will go against their recommendations. The reasons are that
1. The setting is somehow nice, but almost straightforward for someone used to bandits
2. The model assumptions are too strong: an optimal action belongs to the trust set - regret should be redefined especially when this is not the case (Appendix E slightly goes into that direction but keeps the original, unfeasible, benchmark -- hence linear regret). The trust mechanism is also quite simple (and it is immediate to find the trust arms)
3. The regret bound should involve T (the size of the trust arms et) and not K [but this is the case in the proof] and one would expect distribution dependent bounds. I think those are also easily obtainable.
4. I believe the proofs can be simplified a lot; right now they are a lot of cumbersome notations and additional Lemmas. My educated guess is that it could and should be broken down in easier to understand piece.

At the end, although the reviewers were all positive, I am saying again that this is an interesting problem and paper, but I do not see it as in its final version and therefore recommend rejection. Its next revised version will be, I am certain, much better and will certainly be accepted in a future venue.

**Additional Comments On Reviewer Discussion:**

The reviewers were positive, and I was curious about this paper hence I read it myself.

I disagreed with the reviewers on many points (listed above), I believe they did not spot them due to some lack of expertise (they are much more junior than me). This is not a criticism against them, I am 100% sure they gave an honest review, but at the end the expertise of an AC kicks in to make the final decision.

Based on the different weaknesses I have spotted, I could not accept this paper in its current form, I apologize for this, at it must be disappointing for the authors (but they will have the chance to revise and improve their paper)

---

### Decision · Program_Chairs · 2025-01-22

Reject